# COMPLEXITY OF INJECTIVITY AND VERIFICATION OF ReLU NEURAL NETWORKS

## ABSTRACT

Neural networks with ReLU activation play a key role in modern machine learning. Understanding the functions represented by ReLU networks is a major topic in current research as this enables a better interpretability of learning processes.

Injectivity plays a crucial role whenever invertibility of a neural network is necessary, such as, e.g., for inverse problems or generative models. The exact computational complexity of deciding injectivity was recently posed as an open problem (Puthawala et al. [JMLR 2022]). We answer this question by proving coNP-completeness. On the positive side, we show that the problem for a single ReLU-layer is still tractable for small input dimension; more precisely, we present a parameterized algorithm which yields fixed-parameter tractability with respect to the input dimension.

In addition, we study the network verification problem which is of great importance since neural networks are increasingly used in safety-critical systems. We prove that network verification is coNP-hard for a general class of input domains. Our result thus highlights that the hardness of network verification is intrinsic to the ReLU networks themselves, rather than specific input domains.

In this context, we also characterize surjectivity for ReLU networks with one-dimensional output which turns out to be the complement of a basic network verification task. We reveal interesting connections to computational convexity by formulating the surjectivity problem as a zonotope containment problem.

## 1 INTRODUCTION

Neural networks with rectified linear units (ReLUs) are a widely used model in deep learning. In practice, neural networks are trained on finite datasets and are expected to generalize to new, unseen inputs. However, they often exhibit unexpected and erroneous behavior in response to minor input perturbations; see, e.g., Szegedy et al. (2014). Hence, the certification of trained networks is of great importance and necessitates a thorough understanding of essential properties of the function computed by a ReLU network.

Network verification, that is, the question whether for all inputs from a given subset $\mathcal{X}$, the ReLU network outputs a value contained in a given set $\mathcal{Y}$, is a research field gaining high interest recently since neural networks are increasingly used in safety-critical systems like autonomous vehicles (Bojarski et al. (2016)) and collision avoidance for drones (Julian et al. (2019)), see, e.g., Weng et al. (2018); Rössig & Petkovic (2021); Kouvaros & Lomuscio (2021); Katz et al. (2022). Typically, $\mathcal{X}$ and $\mathcal{Y}$ are balls or defined by some linear constraints and the problem is known to be coNP-hard if the sets $\mathcal{X}$ and $\mathcal{Y}$ are part of the input; see Katz et al. (2022); Weng et al. (2018); Sälzer & Lange (2023).

Moreover, recent works focus on studying elementary properties of functions computed by ReLU networks such as injectivity; see, e.g., Puthawala et al. (2022); Haider et al. (2023); Furuya et al. (2023). Injectivity plays a crucial role in many applications where invertibility of a neural network is necessary. Such applications include, for example, generative models, inverse problems, or likelihood estimation; see Puthawala et al. (2022); Furuya et al. (2023) for a more detailed discussion. On the other hand, injectivity might cause privacy issues since the input might be inferred from the output.

Formally, for any number of layers $\ell \in \mathbb{N}$, given weights of a fully connected ReLU network $f \colon \mathbb{R}^d \to \mathbb{R}^m$ with $\ell$ layers, the question is whether the map $f$ is injective. Puthawala et al. (2022) give a mathematical characterization for injectivity of a single ReLU-layer (see Theorem 5), implying an exponential-time algorithm where the exponential part is upper bounded by $\min\{2^m, m^d\}$. In terms of *parameterized complexity*, the problem is *fixed-parameter tractable* with respect to the number $m$ of ReLUs (that is, the superpolynomial part depends only on $m$), and in XP with respect to the input dimension $d$ (that is, polynomial time for constant $d$). The complexity status of deciding injectivity of an $\ell$-layer ReLU neural network (and therefore also, in particular, a single layer), however, is unresolved and posed as an open problem by Puthawala et al. (2022).

Another natural and fundamental property of functions to consider is surjectivity. Hence, to complete the picture alongside injectivity, we initiate the study of surjectivity for ReLU networks, which, to the best of our knowledge, has not been considered before. In fact, we show that deciding surjectivity can be considered as the complement of a certain network verification task.

**Our Contributions.** After some preliminaries in Section 2, we show in Section 3 that deciding injectivity of a ReLU network with $\ell$ layers is coNP-complete (Corollary 8) for any $\ell \in \mathbb{N}$ and thus not polynomial-time solvable, unless P = NP. Notably, our hardness reduction reveals some interesting connections between cut problems in (di)graphs and properties of ReLU networks via graphical hyperplane arrangements. We believe that these connections between seemingly unrelated areas are of independent interest. Moreover, our hardness result implies a running time lower bound of $2^{\Omega(m)}$ for a single ReLU-layer with $m$ neurons based on the Exponential Time Hypothesis (Corollary 7). Hence, the running time $2^m$ is essentially optimal. As regards the input dimension $d$, however, in Section 4 we give an improved algorithm running in $O\big((d+1)^d \cdot \mathrm{poly}\big)$ time. This yields fixed-parameter tractability (Theorem 10) and settles the complexity status for injectivity of a ReLU-layer.

Katz et al. (2022) and Sälzer & Lange (2023) show that the network verification problem is coNP-hard for a single hidden layer. However, their hardness results are based on the unit cube as input set. Thus, it was not clear so far whether there are interesting special cases of input sets on which network verification is solvable in polynomial time. In Section 5, we provide the strongest hardness result for network verification to date (Corollary 13), proving coNP-hardness for every possible input domain $\mathcal{X}$ that contains a ball (of possibly lower dimension under some mild conditions). In particular, our result implies that the computational intractability does not stem from choosing a particular set $\mathcal{X}$, but is rather an intrinsic property of a single hidden ReLU-layer. Since arguably any reasonable input domain contains some ball (this holds, e.g., for all polyhedral sets satisfying the same mild conditions), no special case of verification regarding the input domain is solvable in polynomial time in the worst case. In particular, our result implies hardness for all polyhedra and also all balls as input set, which is not covered by the previous hardness results. As a consequence, one must make (very) specific assumptions on the network in order to obtain tractable cases, e. g., by building special (approximating) networks that are efficiently verifiable (Baader et al. (2020),Wang et al. (2022),Baader et al. (2024)).

Moreover, we give a characterization of surjectivity for ReLU networks with one-dimensional output (Lemma 14) which implies a polynomial-time algorithm for constant input dimension $d$. We then proceed with proving NP-hardness of surjectivity by showing that it can be phrased as (the complement of) a special case of network verification. Finally, we also show that surjectivity can be formulated as a zonotope containment problem, which is of fundamental importance, e.g., in robotics (Kulmburg & Althoff (2021)).

**Related Work.** Puthawala et al. (2022) initiate the study of injectivity of ReLU networks. Their result implicitly yields an exponential-time algorithm. They also mention a connection between injectivity and the *spark* of the weight matrix (that is, the minimum number of linearly dependent rows). Computing the spark is known to be NP-hard (Tillmann & Pfetsch (2014)), and also known not to be fixed-parameter tractable with respect to $d$, unless FPT = W[1] (Panolan et al. (2015)). Haider et al. (2023) use a frame-theoretic approach to study the injectivity of a ReLU-layer on a closed ball. They give an exponential-time algorithm to determine a bias vector for a given weight matrix such that the corresponding map is injective on a closed ball. Furuya et al. (2023) study injectivity of ReLU-layers with linear neural operators.

The network verification problem is known to be coNP-hard for networks with two layers and arbitrary output dimension (Katz et al. (2022) proved NP-hardness for the complement). Sälzer & Lange

(2023) corrected some flaws in the proof and showed coNP-hardness for one-dimensional output. The problem is also known to be coNP-hard to approximate in polynomial time for one-dimensional output if the number of layers is unbounded (Weng et al. (2018)) and for three hidden layers and squashable activation functions Wang et al. (2022). Efficient verifiable networks with heuristics such as the interval bound propagation Gowal et al. (2018) are shown to be universal approximators Wang et al. (2022); Baader et al. (2020), but the networks might have exponential size. On the other hand, obtaining exact representations for all functions computed by ReLU neural networks is not possible such that they are precisely verifiable with interval bound propagation Mirman et al. (2022) or single neuron convex relaxations Baader et al. (2024). Further heuristics include methods such as Deep-Poly (Singh et al. (2019)), DeepZ (Wong et al. (2018)), general cutting planes (Zhang et al. (2024)), multi neuron verification (Ferrari et al. (2022)). Moreover, there exist libraries for verification (Xu et al. (2020); Mao et al. (2024)).

Parameterized complexity has also been studied for the training problem for ReLU neural networks by Froese et al. (2022); Froese & Hertrich (2023). In general, understanding the complexity/expressivity of ReLU neural networks is an important task (Hertrich et al. (2021); Arora et al. (2018)).

## 2 PRELIMINARIES

We introduce the basic definitions and concepts involving ReLU neural networks and their geometry which are relevant for this work.

**Definition 1.** A ReLU-layer with $d$ inputs, $m$ outputs, weights $\mathbf{W} \in \mathbb{R}^{m \times d}$, and biases $\mathbf{b} \in \mathbb{R}^m$ computes a map $\phi_{\mathbf{W},\mathbf{b}} \colon \mathbb{R}^d \to \mathbb{R}^m, \quad \mathbf{x} \mapsto [\mathbf{W}\mathbf{x} + \mathbf{b}]_+$, where $[\cdot]_+ \colon \mathbb{R}^m \to \mathbb{R}^m$ is the *rectifier function* given by $[\mathbf{x}]_+ := (\max\{0, \mathbf{x}_1\}, \dots, \max\{0, \mathbf{x}_m\})$.

For most of our purposes, we can assume $\mathbf{b} = \mathbf{0}$ without loss of generality. In that cases, we omit $\mathbf{b}$ and simplify notation $\phi_{\mathbf{W}} := \phi_{\mathbf{W},\mathbf{0}}$. A deep ReLU network is just a concatenation of compatible ReLU-layers.

**Definition 2.** An $\ell$-layer ReLU network of architecture $(n_0, n_1, \dots, n_{\ell-1}, n_\ell)$ with weights $\mathbf{W}_i \in \mathbb{R}^{n_{i-1} \times n_i}$ and biases $\mathbf{b}_i \in \mathbb{R}^{n_i}$ for $i \in \{1, \dots, \ell\}$ computes a map

$$f \colon \mathbb{R}^d \to \mathbb{R}^m, \quad \mathbf{x} \mapsto \mathbf{W}_\ell \cdot (\phi_{\mathbf{W}_{\ell-1},\mathbf{b}_{\ell-1}} \circ \cdots \circ \phi_{\mathbf{W}_1,\mathbf{b}_1})(\mathbf{x}) + \mathbf{b}_\ell,$$

where $d = n_0$ and $m = n_\ell$.

Whenever we consider an $\ell$-layer ReLU neural network $f \colon \mathbb{R}^d \to \mathbb{R}^m$ as the input to a decision problem in the paper, we implicitly consider its weights and biases as the input. For most of our purposes we can assume without loss of generality $\mathbf{b}_i = \mathbf{0}$ and hence the technical parts only deal with such ReLU-layers.

**Geometry of ReLU-layers.** We review basic definitions from polyhedral geometry; see Schrijver (1986) for more details. For a row vector $\mathbf{w}_i$, the hyperplane $H_{\mathbf{w}_i} := \{\mathbf{x} \in \mathbb{R}^d \mid \mathbf{w}_i \mathbf{x} = 0\}$ subdivides $\mathbb{R}^d$ into half-spaces $H_{\mathbf{w}_i}^+ := \{\mathbf{x} \in \mathbb{R}^d \mid \mathbf{w}_i \mathbf{x} \geq 0\}$ and $H_{\mathbf{w}_i}^- := \{\mathbf{x} \in \mathbb{R}^d \mid \mathbf{w}_i \mathbf{x} \leq 0\}$. A *polyhedron* $P$ is the intersection of finitely many closed halfspaces. A *polyhedral cone* $C \subseteq \mathbb{R}^d$ is a polyhedron such that $\lambda u + \mu v \in C$ for every $u, v \in C$ and $\lambda, \mu \in \mathbb{R}_{\geq 0}$. A hyperplane *supports* $P$ if it bounds a closed halfspace containing $P$, and any intersection of $P$ with such a supporting hyperplane yields a *face* $F$ of $P$. A *polyhedral complex* $\mathcal{P}$ is a finite collection of polyhedra such that (i) $\emptyset \in \mathcal{P}$, (ii) if $P \in \mathcal{P}$ then all faces of $P$ are in $\mathcal{P}$, and (iii) if $P, P' \in \mathcal{P}$, then $P \cap P'$ is a face of both $P$ and $P'$. A *polyhedral fan* is a polyhedral complex in which every polyhedron is a cone. For a matrix $\mathbf{W} \in \mathbb{R}^{m \times d}$, the set of full-dimensional polyhedral cones

$$\mathcal{C}_{\mathbf{W}} := \left\{ \bigcap_{i=1}^m H_{\mathbf{w}_i}^{s_i} \; \middle| \; (s_1, \dots, s_m) \in \{+, -\}^m, \dim\left(\bigcap_{i=1}^m H_{\mathbf{w}_i}^{s_i}\right) = d \right\}$$

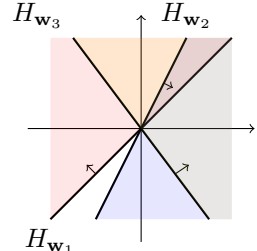

Figure 1: The polyhedral fan induced by an oriented linear hyperplane arrangement given by a matrix $\mathbf{W} \in \mathbb{R}^{3 \times w}$.

subdivides $\mathbb{R}^d$ and the set $\{C \cap H_{\mathbf{w}_i} \mid C \in \mathcal{C}_{\mathbf{W}}, i \in [m]\}$ forms a polyhedral fan. A vector $\mathbf{x}$ in the support $|\Sigma_{\mathbf{W}}| = \bigcup_{\sigma \in \Sigma_{\mathbf{W}}} \sigma$ of the fan $\Sigma_{\mathbf{W}}$ is called

a *breakpoint* of $\phi_{\mathbf{W}}$. See Figure 1 for an illustration of a 2-dimensional polyhedral fan arising from a ReLU-layer.

For $C \in \mathcal{C}_{\mathbf{W}}$, we define the *active* set $I_C := \{j \in [m] \mid \forall \mathbf{x} \in C : \mathbf{w}_j \mathbf{x} \geq 0\}$ and the matrix $\mathbf{W}_C \in \mathbb{R}^{m \times d}$, where $(\mathbf{W}_C)_j := \begin{cases} \mathbf{w}_j, & j \in I_C, \\ \mathbf{0}, & j \in [m] \setminus I_C. \end{cases}$ Note that the map $\phi_{\mathbf{W}}$ is linear on $C$, namely $\phi_{\mathbf{W}}(\mathbf{x}) = \mathbf{W}_C \cdot \mathbf{x}$ for $\mathbf{x} \in C$.

**Geometry of ReLU Neural Networks.** It is well-known that also a deep ReLU neural network $f$ partitions its input space into polyhedra on which $f$ is affine linear (Hanin & Rolnick (2019); Grigsby & Lindsey (2022)). More precisely, let $f_{i,j} = \pi_j \circ \phi_{\mathbf{W}_i, \mathbf{b}_i} \circ \cdots \circ \phi_{\mathbf{W}_1, \mathbf{b}_1}$, where $\pi_j \colon \mathbb{R}^{n_i} \to \mathbb{R}$ is the projection onto the $j$-th coordinate. Then, every $\mathbf{s} = (\mathbf{s}_1, \ldots, \mathbf{s}_{n_{\ell-1}}) \in \{-, +\}^{n_1} \times \ldots \times \{-, +\}^{n_{\ell-1}}$ corresponds to a (possibly empty) polyhedron

$$P_{\mathbf{s}} = \bigcap_{(\mathbf{s}_i)_j = +} \{\mathbf{x} \in \mathbb{R}^d \mid f_{i,j}(\mathbf{x}) \geq 0\} \cap \bigcap_{(\mathbf{s}_i)_j = -} \{\mathbf{x} \in \mathbb{R}^d \mid f_{i,j}(\mathbf{x}) \leq 0\}.$$

The maps $f_{i,j}$ are affine linear on $P_{\mathbf{s}}$ and the coefficients are polynomially bounded in the weights and biases of the neural network. Hence, the polyhedron $P_{\mathbf{s}}$ arises as the intersection of at most $\sum_{i=1}^{\ell} n_{\ell}$ half-spaces whose encoding sizes are polynomially bounded in the weights and biases of $f$. All polyhedra $P_{\mathbf{s}}$ and their faces form a polyhedral complex which we denote by $\Sigma_f$.

A *ray* $\rho$ is a one-dimensional pointed cone; a vector $\mathbf{r}$ is a *ray generator* of $\rho$ if $\rho = \{\lambda \mathbf{r} \mid \lambda \geq 0\}$. For each ray $\rho$ of $\Sigma_f$, let $\mathbf{r}_\rho$ be the unique unit ray generator of $\rho$ having norm 1 and let $\mathcal{R} := \{\mathbf{r}_\rho \mid \rho \text{ is a ray in } \Sigma_f\}$. A ray $\rho \subseteq C$ of a cone $C$ is an *extreme ray* if there do not exist $\lambda_1, \lambda_2 > 0$ and $\rho_1, \rho_2 \subseteq C$ such that $\rho = \lambda_1 \rho_1 + \lambda_2 \rho_2$. We make the following simple observation.

**Observation 3.** *Let $C$ be a pointed polyhedral cone such that $f$ is linear on $C$ and let $\mathbf{r}_1, \ldots, \mathbf{r}_\ell$ be ray generators of the extreme rays of $C$. Then, for each $\mathbf{x} \in C$, there are $\lambda_1, \ldots, \lambda_\ell \in \mathbb{R}$ such that $\mathbf{x} = \sum_{i=1}^{\ell} \lambda_i \mathbf{r}_i$ and $f(\mathbf{x}) = \sum_{i=1}^{\ell} \lambda_i f(\mathbf{r}_i)$.*

Observation 3 implies that the map $f$ is essentially determined by its values on $\mathcal{R}$, if all cones $C \in \Sigma_f$ are pointed.

**(Parameterized) Complexity Theory.** We assume the reader to be familiar with basic concepts from classical complexity theory like P, NP, and NP-completeness. The class coNP contains all decision problems whose complement is in NP. A decision problem is coNP-complete if and only if its complement is NP-complete. Clearly, a coNP-complete problem cannot be solved in polynomial time unless P = NP.

A *parameterized* problem consists of instances $(x, k)$, where $x$ encodes the classical instance and $k \in \mathbb{N}$ is a *parameter*. A parameterized problem is in the class XP if it is polynomial-time solvable for every constant parameter value, that is, in $O(|x|^{f(k)})$ time for an arbitrary function $f$ depending only on the parameter $k$. A parameterized problem is *fixed-parameter tractable* (contained in the class FPT) if it is solvable in $f(k) \cdot |x|^{O(1)}$ time, for an arbitrary function $f$. Clearly, FPT $\subseteq$ XP; see Downey & Fellows (2013) for further details of parameterized complexity.

## 3 coNP-Completeness of Injectivity

In this section we study the computational complexity of $\ell$-ReLU-layer Injectivity, that is, deciding whether a ReLU network $f \colon \mathbb{R}^d \to \mathbb{R}^m$ with $\ell$ layers computes an injective map.

**Proposition 4.** *For every $\ell \in \mathbb{N}$, it holds that $\ell$-ReLU-layer Injectivity is contained in coNP.*

*Proof sketch.* Two maximal polyhedra $P, Q \in \Sigma_f$ serve as a certificate for non-injectivity, since checking whether there are $\mathbf{x}_1 \in P$ and $\mathbf{x}_2 \in Q$ with $\mathbf{x}_1 \neq \mathbf{x}_2$ such that $f(\mathbf{x}_1) = f(\mathbf{x}_2)$ is simply checking feasibility of a linear program. For a rigorous proof one can apply Theorem 3.3 in Sälzer & Lange (2023). $\square$

To show that $\ell$-ReLU-layer Injectivity is coNP-hard for any $\ell \in \mathbb{N}$, it suffices to show that deciding injectivity is already coNP-hard for a single ReLU-layer. Hence, in the remainder of this

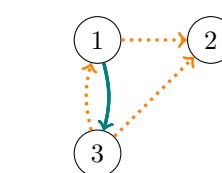 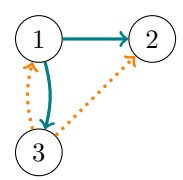 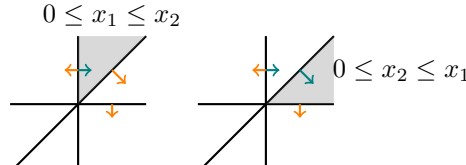

(a) For the ordering $3 \leq 2 \leq 1$ (on the left) and the ordering $3 \leq 1 \leq 2$ (on the right), the arcs that respect the ordering are dotted and colored in orange.

(b) The resulting ReLU network represented by the corresponding oriented hyperplane arrangements. For the gray colored cones, the inactive neurons are colored in orange.

Figure 2: An illustration of the reduction from ACYCLIC 2-DISCONNECTION to RELU-LAYER NON-INJECTIVITY for a digraph with $n = 3$ nodes. An ordering $\pi$ of the coordinates induces an acyclic subset $A_\pi$ of arcs corresponding to inactive neurons on the cone $C_\pi = \{x_{\pi(1)} \leq \cdots \leq x_{\pi(n)}\}$. The active neurons on $C_\pi$ have full rank $n$ if and only if removing the arcs $A_\pi$ results in a weakly connected digraph.

section we study the following decision problem (RELU-LAYER INJECTIVITY): Given a matrix $\mathbf{W} \in \mathbb{R}^{m \times d}$ and a vector $\mathbf{b} \in \mathbb{R}^m$, is the map $\phi_{\mathbf{W},\mathbf{b}}$ injective?

Puthawala et al. (2022, Theorem 2) prove the following characterization of injectivity for a map computed by a single ReLU-layer.

**Theorem 5** (Puthawala et al. (2022)). *A ReLU-layer $\phi_{\mathbf{W}}$ with $\mathbf{W} \in \mathbb{R}^{m \times d}$ is injective if and only if $\mathbf{W}_C$ has (full) rank $d$ for all $C \in \mathcal{C}_{\mathbf{W}}$.*

Since any hyperplane arrangement with $m$ hyperplanes in $d$ dimensions defines at most $O(m^d)$ (also clearly at most $2^m$) cells (Zaslavsky (1975)), Theorem 5 implies an algorithm that solves RELU-LAYER INJECTIVITY in $O(\min\{2^m, m^d\} \cdot \text{poly}(S))$ time, where $S$ denotes the input size.

We prove that RELU-LAYER INJECTIVITY is coNP-complete. To this end, we show NP-completeness for the complement problem RELU-LAYER NON-INJECTIVITY. Based on Theorem 5 and the fact that one can w.l.o.g. assume $\mathbf{b} = \mathbf{0}$ (Puthawala et al., 2022, Lemma 3), the RELU-LAYER NON-INJECTIVITY problem is, given a matrix $\mathbf{W} \in \mathbb{R}^{m \times d}$, to decide if there is a cell $C \in \mathcal{C}_{\mathbf{W}}$ such that $\text{rank}(\mathbf{W}_C) < d$.

For the NP-hardness proof, we reduce from the following directed graph (*digraph*) problem. A digraph $D = (V, A)$ is called *acyclic* if it does not contain oriented cycles and *weakly disconnected* if the underlying graph, i.e., the graph where we have an unoriented edge for every arc, is connected. The ACYCLIC 2-DISCONNECTION problem is, given a digraph $D = (V, A)$, to decide if there is a subset $A' \subseteq A$ of arcs such that $(V, A')$ is acyclic and $(V, A \setminus A')$ is not weakly connected.

The problem is a special case of ACYCLIC $s$-DISCONNECTION where the goal is to remove an acyclic arc set such that the remaining digraph contains at least $s$ weakly connected components. This more general problem is known to be NP-hard if $s$ is part of the input (Figueroa et al. (2017)). In Appendix A.1 we prove NP-hardness for our special case $s = 2$ (Theorem 18). We remark that our reduction implies that ACYCLIC 2-DISCONNECTION cannot be solved in $2^{o(|D|)}$ time unless the Exponential Time Hypothesis[1] fails (Corollary 19).

**Theorem 6.** RELU-LAYER NON-INJECTIVITY *is NP-complete even if every row of $\mathbf{W}$ contains at most two non-zero entries.*

*Proof sketch.* Containment in NP is easy: The set $I_C$ of a cell $C$ with $\text{rank}(\mathbf{W}_C) < d$ serves as a certificate. To prove NP-hardness, we reduce from ACYCLIC 2-DISCONNECTION which is NP-hard by Theorem 18. We sketch the proof here, a detailed proof can be found in Appendix A.1.1. Moreover, the reduction is illustrated in Figure 2. Given a digraph $D = (V, A)$ with $V = [n]$, we construct a ReLU-layer $\phi \colon \mathbb{R}^{n-1} \to \mathbb{R}^{|A|}$ by $\phi(\mathbf{x}) = (\max\{0, \mathbf{x}_i - \mathbf{x}_j\})_{(i,j) \in A}$. For every permutation $\pi \in \mathcal{S}_n$, function $\phi$ is linear on the cone $C_\pi \coloneqq \{\mathbf{x} \mid \mathbf{x}_{\pi(1)} \leq \cdots \leq \mathbf{x}_{\pi(n)}\}$, where we

---

[1]The Exponential Time Hypothesis asserts that 3-SAT cannot be solved in $2^{o(n)}$ time where $n$ is the number of Boolean variables in the input formula (Impagliazzo & Paturi (2001)).

let $\mathbf{x}_n := 0$, since no hyperplane $\mathbf{x}_i = \mathbf{x}_j$ induced by a neuron $\max\{0, \mathbf{x}_i - \mathbf{x}_j\}$ intersects this cone. Let $A_\pi := \{(i,j) \in A \mid \pi(i) \leq \pi(j)\} \subseteq A$ be the (acyclic) subset of arcs respecting the (total) order on the nodes induced by $\pi \in \mathcal{S}_n$. The set of arcs in the complement $A \setminus A_\pi$ is therefore in bijection with the neurons that are active on $C_\pi$, i.e., neurons $\max\{0, \mathbf{x}_i - \mathbf{x}_j\}$ where $\pi(j) \leq \pi(i)$. In Appendix A.1.1, we prove that the digraph $D_\pi = (V, A \setminus A_\pi)$ is weakly connected if and only if

- there is a path in the underlying graph of $D_\pi$ from node $j$ to node $n$, for all $j \in [n-1]$;

- if and only if the standard unit vector $\mathbf{e}_j$ is contained in $\mathrm{span}(\mathbf{W}_{C_\pi})$, for all $j \in [n-1]$;

- if and only if $\mathbf{W}_{C_\pi}$ has full rank.

The illustration for the 2-dimensional case in Figure 2 provides intuition for this chain of equivalent statements. Since the partial order on the nodes induced by an acyclic subset of arcs can be extended to a total order on the nodes, it holds that $\phi$ is not injective if and only if there is $A' \subseteq A$ such that $(V, A')$ is acyclic and $(V, A \setminus A')$ is not weakly connected, proving the correctness of the reduction. $\qquad\square$

The lower bound from Corollary 19 actually transfers to RELU-LAYER INJECTIVITY since our polynomial-time reduction in the proof of Theorem 6 yields a matrix where the number of rows/columns is linear in the number arcs/nodes of the digraph.

**Corollary 7.** RELU-LAYER INJECTIVITY *is coNP-complete even if every row of* $\mathbf{W}$ *contains at most two non-zero entries (and* $\mathbf{b} = \mathbf{0}$*). Moreover,* RELU-LAYER NON-INJECTIVITY *and* RELU-LAYER INJECTIVITY *cannot be solved in* $2^{o(m+d)}$ *time, unless the ETH fails.*

To prove coNP-hardness for an arbitrary number $\ell$ of layers, one can simply reduce RELU-LAYER INJECTIVITY to $\ell$-RELU-LAYER INJECTIVITY by concatenating the layer with a ReLU network with $\ell - 1$ layers that computes the identity map. Hence, we obtain the following corollary.

**Corollary 8.** *For every* $\ell \geq 2$*, it holds that* $\ell$-RELU-LAYER INJECTIVITY *is coNP-complete.*

## 4 AN FPT ALGORITHM FOR RELU-LAYER INJECTIVITY

The complexity results from the previous section exclude polynomial running times as well as running times subexponential in $m + d$ for deciding injectivity for a single ReLU-layer. Nevertheless, we show in this section that the previous upper bound of $m^d$ can be improved to $(d+1)^d$. We achieve this with a branching algorithm which searches for a "non-injective" cell $C$, that is, $\mathbf{W}_C$ has rank strictly less than $d$. The algorithm branches on the ReLUs active in $C$, thus restricting the search space to some cone. The pseudocode is given in Algorithms 1 and 2.

The key idea to bound the running time of our search tree algorithm is to show that there are always at most $d+1$ candidate ReLUs to set active. The candidates are those whose corresponding halfspaces cover the current search space; that is, at least one of them will be active in the sought cell (cf. Line 5 of Algorithm 2). Helly's Theorem (Helly (1923)) ensures that there are always at most $d+1$ such candidates. Using duality of linear programming, we show that these halfspaces can be found in polynomial time. A more detailed proof of the following lemma is given in Appendix A.2.1.

**Lemma 9.** *Let* $C \subseteq \mathbb{R}^d$ *be a cone and let* $\{\mathbf{w}_1, \ldots, \mathbf{w}_n\} \subseteq \mathbb{R}^d$ *be such that* $C$ *is covered by the corresponding half-spaces, that is,* $C \subseteq \bigcup_{i=1}^n H_{\mathbf{w}_i}^+$*. Then, there exists a subset* $A \subseteq [n]$ *of size at most* $d+1$ *computable in polynomial time, such that* $C \subseteq \bigcup_{i \in A} H_{\mathbf{w}_i}^+$*.*

---

**Algorithm 1:** LayerInjectivity

**Input** : $W = \{\mathbf{w}_1, \ldots, \mathbf{w}_m\} \subseteq \mathbb{R}^d$
**Output:** $\mathbf{x} \in \mathbb{R}^d$ with $\mathrm{rank}(\{\mathbf{w}_i \in W \mid \mathbf{w}_i^T \mathbf{x} \geq 0\}) < d$ (if it exists); otherwise, "yes"
1 **if** $\mathrm{rank}(W) < d$, **then return** 0;
2 **if** $\exists \mathbf{x} \in \mathbb{R}^d : \forall i \in [m] : \mathbf{w}_i^T \mathbf{x} < 0$, **then return** $\mathbf{x}$;
3 $x \leftarrow \mathrm{FindCell}(\emptyset, W)$;
4 **if** $x \neq$ "no", **then return** $x$;
5 **return** "yes"

---

---

**Algorithm 2:** FindCell

**Input** : vectors $C = \{\mathbf{c}_1, \ldots, \mathbf{c}_n\} \subseteq \mathbb{R}^d$ and $M = \{\mathbf{m}_1, \ldots, \mathbf{m}_m\} \subseteq \mathbb{R}^d$

**Output:** vector $\mathbf{x} \in \mathbb{R}^d$ (if it exists) such that $\mathbf{c}_i^T \mathbf{x} \geq 0$, for all $i \in [n]$, and
$\text{rank}(C \cup \{\mathbf{m}_i \mid \mathbf{m}_i^T \mathbf{x} \geq 0\}) < d$; otherwise, "no"

1 **if** $\text{rank}(C) = d \vee \{\mathbf{x} \mid \forall i \in [n] : \mathbf{c}_i^T \mathbf{x} \geq 0\} = \{\mathbf{0}\}$, **then return** "no";

2 $I \leftarrow \{i \in \{1, \ldots, m\} \mid \mathbf{m}_i \notin \text{span}(C)\}$;

3 $M \leftarrow \{\mathbf{m}_i \mid i \in I\}$;

4 **if** $\exists \mathbf{x} \in \mathbb{R}^d : \left((\forall i \in [n] : \mathbf{c}_i^T \mathbf{x} \geq 0) \wedge (\forall i \in I : \mathbf{m}_i^T \mathbf{x} < 0)\right)$, **then return** $\mathbf{x}$;

5 compute $A \subseteq I$ such that $\{\mathbf{x} \mid \forall i \in [n] : \mathbf{c}_i^T \mathbf{x} \geq 0\} \subseteq \bigcup_{i \in A} H_{\mathbf{m}_i}^+$ and $|A| \leq d + 1$;

6 **foreach** $i \in A$ **do**

7    $x \leftarrow \text{FindCell}(C \cup \{\mathbf{m}_i\}, M \setminus \{\mathbf{m}_i\})$;

8    **if** $x \neq$ "no", **then return** $x$;

9 **end**

10 **return** "no"

---

As regards the correctness of our algorithm, assume that there exists a vector $\mathbf{x} \in \mathbb{R}^d$ such that $\text{rank}(\{\mathbf{w}_i \mid \mathbf{w}_i^T \mathbf{x} \geq 0\}) = k < d$. Then, after every branching, $\mathbf{x}$ is contained in some cone that is the search space of FindCell, that is, $\mathbf{c}_i^T \mathbf{x} \geq 0$ for all $i \in [n]$. Notice that every branching increases the number of linear independent neurons that are active in the current cone by one. Thus, after at most $k$ branchings, FindCell finds a cone where the map is not injective. Conversely, if the algorithm outputs an $\mathbf{x} \in \mathbb{R}^d$, then the matrix $\mathbf{W}_{\bar{C}}$ corresponding to the cone $\bar{C} := \{\bar{\mathbf{x}} \mid \forall i \in [n] : \mathbf{c}_i^T \bar{\mathbf{x}} \geq 0\}$ containing $\mathbf{x}$ cannot have full rank; otherwise, Line 1 in FindCell would have output "no". Lemma 9 ensures that we can always branch on $d + 1$ neurons. Furthermore, the number of active neurons increases after every branching by one. Hence the search tree has at most $(d + 1)^d$ nodes. We conclude our findings in the following theorem for which we provide a detailed proof in Appendix A.2.2.

**Theorem 10.** RELU-LAYER INJECTIVITY *is solvable in* $O((d + 1)^d \cdot \text{poly}(S))$ *time, where* $S$ *denotes the input size.*

Deciding injectivity for a deep neural network $f = \phi_\ell \circ \cdots \circ \phi_1$ is even more involved. Clearly, if all layer maps $\phi_i$ are injective, then also $\Phi$ is injective. The converse does not hold, however, since $\phi_i$ only needs to be injective on the image $(\phi_{i-1} \circ \cdots \circ \phi_1)(\mathbb{R}^d)$ for all $i \in [\ell] := \{1, \ldots, \ell\}$. Hence, it is unclear whether the problem is still contained in FPT when parameterized by $d$.

## 5   VERIFICATION AND SURJECTIVITY

In this section, we study the network verification task and prove hardness for a very general class of input sets. To this end, we call a sequence $S = (S_d)_{d \in \mathbb{N}}$ of subsets $S_d \subseteq \mathbb{R}^d$ *reasonable* if there exists an algorithm that on input $k \in \mathbb{N}$ computes in $\text{poly}(k)$ time a $d \in \mathbb{N}, \mathbf{z} \in \mathbb{R}^d$ and a $k$-dimensional affine space $A$ in $\mathbb{R}^d$ (given by a basis and a translation vector) such that there is a $k$-dimensional ball with center $\mathbf{z}$ contained in $S_d$ that affinely spans $A$. For a sequence $S = (S_d)_{d \in \mathbb{N}}$ of reasonable sets and a polyhedron $Q_{\mathbf{t}} := \{\mathbf{x} \in \mathbb{R}^m \mid \mathbf{x}_i \leq \mathbf{t}_i \text{ for all } i \in [m]\}$ where $\mathbf{t} \in \mathbb{R}^m$, the task to decide for a given $\ell$-layer ReLU network $f : \mathbb{R}^d \to \mathbb{R}^m$ whether $f(S_d) \subseteq Q_{\mathbf{t}}$ we call $\ell$-LAYER RELU $(S, Q_{\mathbf{t}})$-VERIFICATION.

Even though the definition of a reasonable sequence is a bit involved, they include a wide range of set-sequences. For example, they trivially include all sequences of sets that contain an open neighborhood of the origin and therefore in particular all sequences of full-dimensional polyhedra containing the origin in the interior. This shows, in comparison to previous existing coNP-hardness proofs, that the hardness of verification does not rely on the complexity of the polyhedron in the input domain but is intrinsic to the ReLU networks themselves.

To show that the problem is coNP-hard, it suffices to show hardness for one hidden layer and one-dimensional output and hence in the remaining part we study the following decision problem for an arbitrary $t \in \mathbb{R}$.

2-LAYER ReLU $(S, t)$-VERIFICATION

**Input:**    Matrices $\mathbf{W}_1 \in \mathbb{R}^{n \times d}, \mathbf{W}_2 \in \mathbb{R}^{1 \times n}, \mathbf{b}_1 \in \mathbb{R}^n$, and $b_2 \in \mathbb{R}$.
**Question:** Is $\mathbf{W}_2 \cdot \phi_{\mathbf{W}_1, \mathbf{b}_1}(\mathbf{x}) + b_2 \le t$ for all $\mathbf{x} \in S_d$?

We will see in Theorem 12 that the hardness of 2-LAYER ReLU $(S, t)$-VERIFICATION stems from the hardness of deciding whether a ReLU network computes a map that attains a positive output value. Thus, we will first show that the 2-LAYER ReLU POSITIVITY problem is NP-complete: Given Matrices $\mathbf{W}_1 \in \mathbb{R}^{n \times d}, \mathbf{W}_2 \in \mathbb{R}^{1 \times n}$, is there an $\mathbf{x} \in \mathbb{R}^d$ such that $\mathbf{W}_2 \cdot \phi_{\mathbf{W}_1}(\mathbf{x}) > 0$?

Afterwards, we give a reduction from the complement of 2-LAYER ReLU POSITIVITY to 2-LAYER ReLU $(S, t)$-VERIFICATION implying that 2-LAYER ReLU $(S, t)$-VERIFICATION and thus $\ell$-LAYER ReLU $(S, Q_{\mathbf{t}})$-VERIFICATION are coNP-hard.

## 5.1 NP-COMPLETENESS OF 2-LAYER ReLU POSITIVITY

We prove NP-completeness of 2-LAYER ReLU POSITIVITY via reduction from POSITIVE CUT. For a graph $G = (V, E)$, we denote by $E(S, V \setminus S) \coloneqq \{\{u, v\} \in E \mid u \in S, v \notin S\}$ the edges in the cut induced by $S \subseteq V$. The POSITIVE CUT problem is to decide for a given graph $G = (V, E)$ with edge weights $w \colon E \to \mathbb{Z}$ whether there is a subset $S \subseteq V$ such that $\sum_{e \in E(S, V \setminus S)} w(e) > 0$, which we prove to be NP-complete in Appendix A.3.4. A detailed proof of the following theorem is given in Appendix A.3.5.

**Theorem 11.** 2-LAYER ReLU POSITIVITY *is NP-complete.*

*Proof sketch.* The problem is contained in NP since, by Observation 3, it is sufficient to identify an extreme ray $\rho$ as a certificate for positivity.

To prove NP-hardness, we reduce from POSITIVE CUT. Given a weighted graph $(G = (V, E), w)$, we define a 2-layer ReLU neural network $f \colon \mathbb{R}^{|V|} \to \mathbb{R}$, where we have two hidden neurons for every $e \in E$ and the output weight of the neurons are the corresponding weights of the edges. More precisely,

$$f(\mathbf{x}) = \sum_{\{i, j\} \in E} w(\{i, j\}) \cdot ([\mathbf{x}_i - \mathbf{x}_j]_+ + [\mathbf{x}_j - \mathbf{x}_i]_+).$$

The weights of the cuts are now stored as function values on certain vectors. For a subset $S \subseteq V$, let $\mathbf{r}_S \coloneqq \sum_{i \in S} \mathbf{e}_i \in \mathbb{R}^{|V|}$ and $\mathbf{r}'_S \coloneqq -\sum_{i \in S \setminus V} \mathbf{e}_i \in \mathbb{R}^d$. It follows easily that

$$f(\mathbf{r}_S) = f(\mathbf{r}'_S) = \sum_{\{i, j\} \in E(S, V \setminus S)} w(\{i, j\}).$$

We proceed by showing that every $\mathbf{x}$ is the conic combination of such $\mathbf{r}_S$ and $\mathbf{r}'_S$. Hence, there is an $\mathbf{x} \in \mathbb{R}^{|V|}$ such that $f(\mathbf{x}) > 0$ if and only if there is an $S \subseteq V$ such that $f(\mathbf{r}_S) = \sum_{e \in E(S, V \setminus S)} w(e) > 0$, proving the correctness of the reduction. $\quad\square$

## 5.2 coNP-HARDNESS OF VERIFICATION

We continue with the coNP-hardness for network verification using the NP-hardness of 2-LAYER ReLU POSITIVITY.

**Theorem 12.** *For every reasonable set-sequence $S$ and $t \in \mathbb{R}$, it holds that* 2-LAYER ReLU $(S, t)$-VERIFICATION *is coNP-hard.*

*Proof sketch.* In Appendix A.3.1, we provide a reduction from the complement of 2-LAYER ReLU POSITIVITY that we sketch here. Let $f \colon \mathbb{R}^k \to \mathbb{R}$ be the function computed by an instance of 2-LAYER ReLU POSITIVITY and let the dimension $d$, the affine space $A$ and the point $\mathbf{z} \in \mathbb{R}^d$ be the output of the algorithm that exists due to the fact that $(S_d)_{d \in \mathbb{N}}$ is a reasonable sequence of sets. Let $T \colon \mathbb{R}^d \to \mathbb{R}^d$ be the affine map that maps $\mathbf{z}$ to the origin composed with the affine map that projects orthogonal to $A$ and afterwards maps isomorphic to $\mathbb{R}^k$. Then we prove that $\mathbf{x} \mapsto (f \circ T)(\mathbf{x}) + t$ is the function computed by a "no"-instance of 2-LAYER ReLU $(S, t)$-VERIFICATION if and only if $f$ attains a positive output value. $\quad\square$

To prove coNP-hardness for an arbitrary number $\ell \geq 2$ of layers and arbitrary output dimension $m$, one can simply provide a reduction from 2-LAYER RELU $(S, t)$-VERIFICATION to $\ell$-LAYER RELU $(S, Q_{\mathbf{t}})$-VERIFICATION by concatenating the 2-layer ReLU network with a ReLU network with $\ell - 1$ layers that computes $m$ many maps in parallel that translate $t$ to $\mathbf{t} \in \mathbb{R}^m$ coordinate-wise. Hence, we obtain the following corollary.

**Corollary 13.** *For every reasonable set-sequence $S$, every polyhedron $Q_{\mathbf{t}} \subseteq \mathbb{R}^m$ with $\mathbf{t} \in \mathbb{R}^m$ and every $\ell \geq 2$, it holds that $\ell$-LAYER RELU $(S, Q_{\mathbf{t}})$-VERIFICATION is coNP-hard.*

## 5.3 SURJECTIVITY

In this section, we study the task to decide whether a given ReLU network $f \colon \mathbb{R}^d \to \mathbb{R}$ with $\ell$ layers computes a surjective map. We will call this task $\ell$-LAYER RELU SURJECTIVITY. We start with some simple observations characterizing surjectivity; see Appendix A.3.2 for a proof.

**Lemma 14.** *For a ReLU network $f = \mathbf{W}_\ell \circ \phi_{\mathbf{W}_{\ell-1}, \mathbf{b}_{\ell-1}} \circ \cdots \circ \phi_{\mathbf{W}_1, \mathbf{b}_1} + b_\ell$ we denote by $f_0 \coloneqq \mathbf{W}_\ell \circ \phi_{\mathbf{W}_{\ell-1}} \circ \cdots \circ \phi_{\mathbf{W}_1}$ the corresponding ReLU network without biases. Then the following holds:*

a) *$f_0$ is surjective if and only if there are $\mathbf{v}^+, \mathbf{v}^- \in \mathbb{R}^d$ such that $f_0(\mathbf{v}^+) > 0$ and $f_0(\mathbf{v}^-) < 0$.*

b) *$f$ is surjective if and only if $f_0$ is surjective*

c) *The map $f$ is surjective if and only if there exist two ray generators $\mathbf{r}^+, \mathbf{r}^-$ of two rays $\rho^+, \rho^- \in \Sigma_{f_0}$ such that $f_0(\mathbf{r}^+) > 0$ and $f_0(\mathbf{r}^-) < 0$.*

Lemma 14 b) implies that we can assume that the ReLU network has no biases without loss of generality. Furthermore, Lemma 14 c) implies an exponential-time algorithm for $\ell$-LAYER RELU SURJECTIVITY, since $\Sigma_{f_0}$ contains at most $O\left(\binom{(\prod_{i=1}^\ell n_i)^d}{d-1}\right)$ many rays. In particular, for $\ell = 2$, the fan $\Sigma_{\mathbf{W}_1}$ contains at most $\min\{2n_1^{d-1}, 2^{n_1+1}\}$ rays; that is, the problem is in XP when parameterized by $d$ and in FPT when parameterized by $n_1$. Moreover, since two ray generators $\mathbf{r}^+, \mathbf{r}^-$ of two rays $\rho^+, \rho^- \in \Sigma_{f_0}$ serve as a certificate, we obtain the following proposition.

**Proposition 15.** *For every $\ell \in \mathbb{N}$, it holds that $\ell$-LAYER RELU SURJECTIVITY is in NP.*

Again, to show that the problem is NP-hard, it suffices to show hardness for one hidden layer and one-dimensional output. Hence, in the remainder we study 2-LAYER RELU SURJECTIVITY, where we have one hidden layer with $n$ neurons (with weights $\mathbf{W}_1 \in \mathbb{R}^{n \times d}$ and bias $\mathbf{b}_1 \in \mathbb{R}^n$) with ReLU activation, and an output layer with one output neuron (with weights $\mathbf{W}_2 \in \mathbb{R}^{1 \times n}$) without activations. The network then computes the map $f \colon \mathbb{R}^d \to \mathbb{R}$ with $f(\mathbf{x}) \coloneqq \mathbf{W}_2 \cdot \phi_{\mathbf{W}_1, \mathbf{b}}(\mathbf{x})$.

In fact, to decide surjectivity, it is actually enough to find one (say the positive) ray generator since it is easy to find some point $\mathbf{x}$ where $f$ is non-zero (we argue below that w.l.o.g. $f(\mathbf{x}) < 0$). To find $\mathbf{x}$, we choose an arbitrary full-dimensional cone and determine whether $\mathbf{W}_C$ is the zero map or not. In the former case, one can pick an arbitrary full-dimensional cone $C'$ that shares a facet with $C$. Then, $\mathbf{W}_{C'} \neq \mathbf{0}$ since the neuron defining the facet must be active in $C'$. Hence, one finds a point $\mathbf{x} \in C'$ with $f(\mathbf{x}) \neq 0$. We give a more detailed proof of the following lemma in Appendix A.3.3.

**Lemma 16.** *One can check in polynomial time whether $f = 0$, and otherwise find a point $\mathbf{x}^* \in \mathbb{R}^d$ such that $f(\mathbf{x}^*) \neq 0$.*

Lemma 16 implies that 2-LAYER RELU SURJECTIVITY is polynomially equivalent to 2-LAYER RELU POSITIVITY (if $f(\mathbf{x}^*) > 0$, then replace $\mathbf{W}_2$ with $-\mathbf{W}_2$ such that $f(\mathbf{x}^*) < 0$) and hence we obtain the following corollary. Moreover,

**Corollary 17.** *For every $\ell \geq 2$, it holds that $\ell$-LAYER RELU SURJECTIVITY is NP-complete.*

Clearly, the above result implies NP-hardness for the general case where the output dimension $m$ is part of the input. It is unclear, however, whether containment in NP also holds for larger output dimension.[2]

---

[2]We believe that the problem might be $\Pi_2^{\mathrm{p}}$-complete for $m \geq 2$.

## 5.4 ZONOTOPE FORMULATION

We conclude this section with an alternative formulation of 2-LAYER RELU POSITIVITY based on a duality of convex piecewise linear functions and polytopes. Interestingly, this yields a close connection to zonotope problems which arise in areas such as robotics and control. Let $\mathcal{F}_d$ be the set of convex piecewise linear functions from $\mathbb{R}^d$ to $\mathbb{R}$ and let $\mathcal{P}_d$ be the set of polytopes in $\mathbb{R}^d$. For every $f \in \mathcal{F}_d$, there are $\{\mathbf{a}_i \in \mathbb{R}^d\}_{i \in I}$ such that $f(\mathbf{x}) = \max_{i \in I}\{\mathbf{a}_i^T \mathbf{x}\}$ and there is a bijection $\varphi \colon \mathcal{F}_d \to \mathcal{P}_d$ given by $\varphi\big(\max_{i \in I}\{\mathbf{a}_i^T \mathbf{x}\}\big) = \text{conv}\,\{\mathbf{a}_i \mid i \in I\}$ where the inverse is the *support function* $\varphi^{-1} \colon \mathcal{P}_d \to \mathcal{F}_d$ given by $\varphi^{-1}(P)(\mathbf{x}) = \max\big\{\langle \mathbf{x}, \mathbf{y}\rangle \mid \mathbf{y} \in P\big\}$.

Furthermore, $\varphi$ is a semi-ring isomorphism between the semi-rings $(\mathcal{F}_d, \max, +)$ and $(\mathcal{P}_d, \text{conv}, +)$, where $+$ is either the pointwise addition or the Minkowski sum, respectively.[3]

A *zonotope* is a Minkowski sum of line segments, i.e., given a matrix $\mathbf{G} \in \mathbb{R}^{n \times d}$, the corresponding zonotope is given by $Z(\mathbf{G}) \coloneqq \Big\{\mathbf{x} \in \mathbb{R}^d \mid \mathbf{x} \in \sum_{i \in [n]} \text{conv}\{\mathbf{0}, \mathbf{g}_i\}\Big\}$.

We now show that 2-LAYER RELU POSITIVITY is equivalent to deciding (non-)containment of certain zonotopes. Given the map $f(\mathbf{x}) = \mathbf{W}_2 \cdot \phi_{\mathbf{W}_1}(\mathbf{x})$ (where $\mathbf{W}_2 \in \{-1, 1\}^{1 \times n}$), we define the sets $I^+ \coloneqq \{i \in [n] \mid (\mathbf{W}_2)_i = 1\}$ and $I^- \coloneqq [n] \setminus I^+$ and let $\mathbf{W} \coloneqq \mathbf{W}_1$. Note that we have

$$f(\mathbf{x}) = \sum_{i \in I^+} \max\{0, \mathbf{w}_i \mathbf{x}\} - \sum_{i \in I^-} \max\{0, \mathbf{w}_i \mathbf{x}\},$$

and therefore $f = \varphi^{-1}(Z^+) - \varphi^{-1}(Z^-)$, where $Z^+ \coloneqq \sum_{i \in I^+} \text{conv}\{\mathbf{0}, \mathbf{w}_i\} = Z(\mathbf{W}_{I^+})$, and $Z^- \coloneqq \sum_{i \in I^-} \text{conv}\{\mathbf{0}, \mathbf{w}_i\} = Z(\mathbf{W}_{I^-})$. Note that the support functions $\varphi^{-1}(Z^+)$ and $\varphi^{-1}(Z^-)$ can only attain nonnegative values and $Z^+ \subseteq Z^-$ implies that $\varphi^{-1}(Z^+) \le \varphi^{-1}(Z^-)$. Moreover, if for a $\mathbf{v} \in Z^+$, it holds that $\varphi^{-1}(Z^+)(\mathbf{v}) > \varphi^{-1}(Z^-)(\mathbf{v})$, then $\mathbf{v} \notin Z^-$. Therefore, there exists a $\mathbf{v} \in \mathbb{R}^d$ such that $\varphi^{-1}(Z^+)(\mathbf{v}) > \varphi^{-1}(Z^-)(\mathbf{v})$ if and only if $Z^+ \not\subseteq Z^-$.

Notably, Kulmburg & Althoff (2021) already showed that ZONOTOPE CONTAINMENT (that is, the question whether $Z^+ \subseteq Z^-$) is coNP-hard, which implies our NP-hardness of 2-LAYER RELU POSITIVITY (Theorem 11). Nevertheless, we believe that our reduction is more accessible and direct and provides a different perspective on the computational hardness. The question whether ZONOTOPE CONTAINMENT is fixed-parameter tractable with respect to $d$ is, to the best of our knowledge, open.

## 6 CONCLUSION

We showed the strongest hardness result for network verification known so far and thereby excluded polynomial-time algorithms (in the worst case) for almost all restrictions on the input set. Moreover, we initiate the complexity-theoretic study of deciding the two elementary properties injectivity and surjectivity for functions computed by ReLU networks. We exclude polynomial-time algorithms for solving both problems, and prove fixed-parameter tractability for injectivity of a single layer. It turned out that surjectivity is a special case of network verification and is also equivalent to zonotope containment. Our results build new bridges between seemingly unrelated areas, and yield new insights into the complexity and expressiveness of ReLU neural networks. We close with some open questions:

- Can the running time for RELU-LAYER INJECTIVITY be improved? Or is it possible to prove a lower bound of $2^{\Omega(d \log d)}$?
- What is the (parameterized) complexity of deciding injectivity for a ReLU neural network with two hidden layers?
- Is 2-LAYER RELU SURJECTIVITY (or equivalently ZONOTOPE CONTAINMENT) fixed-parameter tractable with respect to the input dimension $d$?
- How can surjectivity be characterized for output dimension $m \ge 2$? What is the complexity of the decision problem?
- What is the complexity of bijectivity for 2-layer ReLU networks?

---

[3]See, e.g., Zhang et al. (2018) for more details on this correspondence.

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

## A    APPENDIX

### A.1    APPENDIX TO SECTION 3

To prove that ACYCLIC 2-DISCONNECTION is NP-hard, we reduce from the following problem.

3-UNIFORM HYPERGRAPH 2-COLORING

**Input:**    A 3-uniform hypergraph $H = (V, E)$, that is, $|e| = 3$ for all $e \in E$.
**Question:**    Is there a 2-coloring of the nodes $V$ such that no hyperedge is monochromatic?

**Theorem 18.** ACYCLIC 2-DISCONNECTION *is NP-hard.*

*Proof.* We give a reduction from 3-UNIFORM HYPERGRAPH 2-COLORING which is NP-hard Lovász (1973). Let $H = (V, E)$ be a 3-uniform hypergraph with $|V| = n$ and $|E| = m$. We construct a digraph $D = (U, A)$ as follows: For each $i \in \{0, 1\}$, we define a node set $U_i$ with the $n + 2m$ nodes

$$U_i := \{v_i \mid v \in V\} \cup \bigcup_{e \in E} \{e_i, e_i'\}.$$

Moreover, for each node $v \in V$, we define a node set $X_v$ with the $2 \deg(v) + 2$ nodes

$$X_v := \{x_v, x'_v\} \cup \{x_{v,e,0}, x_{v,e,1} \mid e \in E, v \in e\}.$$

Finally, we define the node set $Q := \{q_{e,i}, q'_{e,i} \mid e \in E, i \in \{0,1\}\}$ and let

$$U := U_0 \cup U_1 \cup \bigcup_{v \in V} X_v \cup Q.$$

The arc set $A$ is defined as follows: For $i \in \{0,1\}$, we connect the nodes in $U_i$ with $2(|U_i| - 1)$ arcs to a strongly connected path, that is, for an arbitrary ordering $b_1, \ldots, b_{|U_i|}$ of the nodes in $U_i$, we insert the arcs $(b_j, b_{j+1})$ and $(b_{j+1}, b_j)$ for each $j \in [|U_i| - 1]$. Analogously, we also connect all nodes in $X_v$ for each $v \in V$ to a strongly connected path. Moreover, for each $v \in V$, we insert the cyclic arcs $(v_0, x_v)$, $(x_v, v_1)$, $(v_1, x'_v)$, and $(x'_v, v_0)$. Finally, for each hyperedge $e = \{u, v, w\} \in E$ and each $i \in \{0,1\}$, we insert the arcs

- $(q_{e,i}, q'_{e,i})$ and $(q'_{e,i}, q_{e,i})$,
- $(x_{u,e,i}, e_i)$, $(e_i, q_{e,i})$, $(q_{e,i}, e'_i)$, and $(e'_i, x_{u,e,i})$, and
- $(x_{v,e,i}, q_{e,i})$, $(q_{e,i}, x_{w,e,i})$, $(x_{w,e,i}, q'_{e,i})$, and $(q'_{e,i}, x_{v,e,i})$.

Overall, the constructed digraph $D$ contains $O(n + m)$ many nodes and arcs.

For the correctness, assume first that there is a 2-coloring of the nodes of $H$ such that no hyperedge is monochromatic and let $V_i \subseteq V$ denote the set of nodes with color $i$. We construct a solution $A' \subseteq A$ for $D$ as follows: For each $v \in V_0$, $A'$ contains the arcs $(x_v, v_1)$ and $(v_1, x'_v)$, and for each $v \in V_1$, $A'$ contains the arcs $(v_0, x_v)$ and $(x'_v, v_0)$. Clearly, these arcs are acyclic. Further, consider a hyperedge $e = \{u, v, w\} \in E$. Since $e$ is not monochromatic, it follows that exactly one of its nodes is colored with one of the two colors, say 0 (the other case is analogous), and the two other nodes have color 1. If $u$ is colored 0, then $A'$ contains the arcs $(x_{u,e,1}, e_1)$, $(e'_1, x_{u,e,1})$, $(e_0, q_{e,0})$, and $(q_{e,0}, e'_0)$. If $v$ has color 0 (the case where $w$ has color 0 is analogous), then $A'$ contains the arcs $(x_{u,e,0}, e_0)$, $(e'_0, x_{u,e,0})$, $(x_{v,e,1}, q_{e,1})$, $(q_{e,1}, x_{v,e,1})$, $(x_{w,e,0}, q'_{e,0})$, and $(q_{e,0}, x_{w,e,0})$. It can easily be verified that $A'$ is acyclic. Moreover, $(U, A \setminus A')$ is not weakly connected since e.g. all nodes in $X_v$ with $v \in V_0$ are disconnected from all nodes in $U_1$.

Conversely, assume that there is a solution $A' \subseteq A$ for $D$. First, observe that in $D' := (U, A \setminus A')$ all nodes in $U_0$ are weakly connected (since they are strongly connected in $D$). The same holds for $U_1$ and for each $X_v$, $v \in V$. Moreover, for each $v \in V$, the set $X_v$ is weakly connected to exactly one of the sets $U_0$ or $U_1$. To see this, note that $X_v$ cannot be disconnected from both $U_0$ and $U_1$ due to the cycle involving the nodes $x_v$ and $x'_v$. It follows that also no $X_v$ can be weakly connected to both $U_0$ and $U_1$ since then $D'$ would be weakly connected because also each node in $Q$ is connected to some $X_v$ due to its cyclic connections. Hence, we assign each node $v \in V$ the color $i \in \{0,1\}$ if and only if $X_v$ is weakly connected to $U_i$.

It remains to check that each hyperedge $e = \{u, v, w\} \in E$ is not monochromatic. Assume the contrary, that is, $X_u$, $X_v$, and $X_w$ are weakly connected to (wlog) $U_0$. Then, by construction, also $q_{e,1}$ and $q'_{e,1}$ are weakly connected to $U_0$. Since $X_u$ is not weakly connected to $U_1$ and therefore $A'$ contains $(x_{u,e,1}, e_1)$ and $(e'_1, x_{u,e,1})$, it follows that $e_1$ is weakly connected to $q_{e,1}$, since otherwise $A'$ would not be acyclic. Therefore, $e_1$ is also weakly connected to $U_0$, which yields a contradiction since then $D'$ would be weakly connected. $\square$

We remark that our reduction implies a running time lower bound based on the Exponential Time Hypothesis[4] (ETH). As discussed by Kratsch & Le (2016), there is no algorithm solving a 3-UNIFORM HYPERGRAPH 2-COLORING-instance $(V, E)$ in $2^{o(|E|)}$ time assuming ETH. Our polynomial-time reduction in the proof of Theorem 18 constructs a digraph $D$ of size $O(|V| + |E|)$. Notice that $O(|V| + |E|) \subseteq O(|E|)$ since we can assume $|V| \leq 3|E|$ (isolated nodes can trivially be removed). Hence, any algorithm solving ACYCLIC 2-DISCONNECTION in $2^{o(|D|)}$ time would imply a $2^{o(|E|)}$-time algorithm for 3-UNIFORM HYPERGRAPH 2-COLORING.

**Corollary 19.** ACYCLIC 2-DISCONNECTION *cannot be solved in* $2^{o(|D|)}$ *time unless the ETH fails.*

---

[4]The Exponential Time Hypothesis asserts that 3-SAT cannot be solved in $2^{o(n)}$ time where $n$ is the number of Boolean variables in the input formula (Impagliazzo & Paturi (2001)).

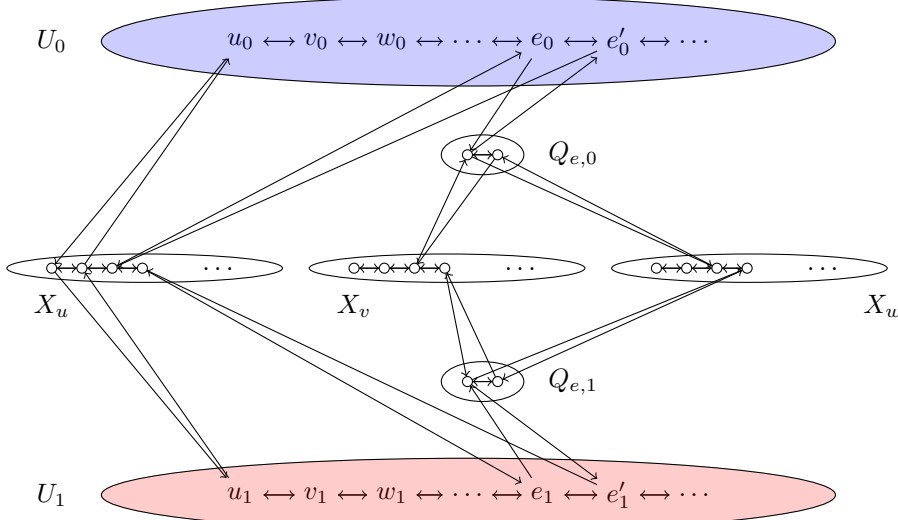

Figure 3: The encoding of one hyperege $\{u, v, w\} \in E$ in the digraph $D$. The cyclic directed paths connecting $X_v$ respectively $X_w$ with $U_0$ and $U_1$ are not drawn in order to not overload the figure.

### A.1.1 PROOF OF THEOREM 6

*Proof.* Containment in NP is easy: The set $I_C$ of a cell $C$ with $\text{rank}(\mathbf{W}_C) < d$ serves as a certificate.

For the NP-hardness, we reduce from ACYCLIC 2-DISCONNECTION which is NP-hard by Theorem 18. For a given digraph $D = (V = \{v_1, \ldots, v_n\}, A = \{a_1, \ldots, a_m\})$, we construct the matrix $\mathbf{W} \in \mathbb{R}^{m \times (n-1)}$ as follows: For every arc $a = (v_i, v_j) \in A$, we add the row vector $\mathbf{w}_a \in \mathbb{R}^{n-1}$ with

$$(\mathbf{w}_a)_\ell = \begin{cases} 1, & \ell = i \\ -1, & \ell = j \\ 0, & \text{otherwise} \end{cases} .$$

For the correctness, assume first that there is a solution $A' \subseteq A$ for $D$. Let $I' := \{i \mid a_i \in A'\} \subseteq [m]$. We claim that there is a cell $C \in \mathcal{C}_\mathbf{W}$ with $I_C \subseteq [m] \setminus I'$. To see this, let $v_{\pi_1}, v_{\pi_2}, \ldots, v_{\pi_n}$ be a topological ordering of $V$ such that all arcs in $A'$ point "from left to right". Such an ordering exists since $A'$ is acyclic. Let $\pi_k = n$ and let $\mathbf{x} \in \mathbb{R}^{n-1}$ be such that

$$\mathbf{x}_{\pi_1} < \cdots < \mathbf{x}_{\pi_{k-1}} < 0 < \mathbf{x}_{\pi_{k+1}} < \cdots < \mathbf{x}_{\pi_n}.$$

Then, no neuron $i \in I'$ corresponding to an arc $a_i = (v_j, v_\ell) \in A'$ is active at $\mathbf{x}$ since

$$\mathbf{w}_{a_i}\mathbf{x} = \begin{cases} \mathbf{x}_j - \mathbf{x}_\ell, & j \neq n \wedge \ell \neq n \\ -\mathbf{x}_\ell, & j = n \\ \mathbf{x}_j, & \ell = n \end{cases} < 0.$$

Hence, $\mathbf{x}$ is contained in some cell $C$ with $I_C \subseteq [m] \setminus I'$. Now, since $(V, A \setminus A')$ is not weakly connected, it follows that $\text{rank}(\mathbf{W}_C) < n - 1$. To see this, note that there must be a node $v_i \in V$ that is not weakly connected to $v_n$, that is, there is no undirected path from $v_i$ to $v_n$. If $\text{rank}(\mathbf{W}_C) = n - 1$, then there exists a linear combination $\sum_{j \in I_C} c_j \mathbf{w}_{a_j} = \mathbf{e}_i$ of the $i$-th unit vector in $\mathbb{R}^{n-1}$. But this implies the existence of an undirected path from $v_i$ to $v_n$ corresponding to some arcs in $\{a_j \mid c_j \neq 0\}$, which yields a contradiction. To see this, note first that $(\mathbf{e}_i)_i = 1$ implies that $c_j \neq 0$ for some $j \in I_C$ such that $(\mathbf{w}_{a_j})_i \neq 0$. Clearly, $a_j = (v_i, v_n)$ or $a_j = (v_n, v_i)$ is not possible. Hence, $a_j$ must be an arc between $v_i$ and some $v_k \neq v_n$. But then, we have $(c_j \mathbf{w}_{a_j})_k \neq 0$, whereas $(\mathbf{e}_i)_k = 0$. Therefore, there exists another $j' \in I_C$ such that $c_{j'} \neq 0$ and $(\mathbf{w}_{a_{j'}}) \neq 0$. Again, it is not possible that $a_{j'} = (v_k, v_n)$ or $a_{j'} = (v_n, v_k)$. However, since $I_C$ is finite, repeating this argument yields a contradiction.

For the reverse direction, let $P \in \mathcal{C}_\mathbf{W}$ be a cell with $\text{rank}(\mathbf{W}_C) < n - 1$. Then, there exists a point $\mathbf{x} \in P$. Now, let $i \in [m] \setminus I_C$ be a neuron corresponding to arc $a_i = (v_j, v_\ell)$ that is not active

at $\mathbf{x}$. Then, this implies that $\mathbf{x}_j - \mathbf{x}_\ell < 0$ if $j < n$ and $\ell < n$. If $j = n$, then this implies $\mathbf{x}_\ell > 0$ and if $\ell = n$, then this implies $\mathbf{x}_j < 0$. Hence, $(V, A')$ with $A' := \{a_i \mid i \in [m] \setminus I_C\}$ is acyclic since any cycle would lead to a contradiction. Now, we claim that $(V, A \setminus A')$ cannot be weakly connected. Otherwise, there exists an undirected path from each $v_i$, $i \in [n-1]$ to $v_n$. Let $v_i = v_{j_0}, v_{j_1}, \ldots, v_{j_t} = v_n$ be the nodes of such a path along the arcs $a_{\ell_1}, \ldots, a_{\ell_t}$. Then, there exists the linear combination $\sum_{k=1}^{t} c_k \mathbf{w}_{a_{\ell_k}} = \mathbf{e}_i$, where $c_k := 1$ if $a_{\ell_k} = (v_{j_{k-1}}, v_{j_k})$ and $c_k := -1$ if $a_{\ell_k} = (v_{j_k}, v_{j_{k-1}})$. But this implies $\text{rank}(\mathbf{W}_C) = n - 1$, which yields a contradiction. $\qquad\square$

### A.2   APPENDIX TO SECTION 4

#### A.2.1   PROOF OF LEMMA 9

*Proof.* We use strong duality of linear programming. To that end, let $\mathbf{A} \in \mathbb{R}^{m \times d}$ such that $C = \{\mathbf{x} \in \mathbb{R}^d \mid \mathbf{A}\mathbf{x} \geq \mathbf{0}\}$ and let $\mathbf{W} := (\mathbf{w}_1, \ldots, \mathbf{w}_n)^T \in \mathbb{R}^{n \times d}$. Since $C \subseteq \bigcup_{i=1}^{n} H_{\mathbf{w}_i}^+$, it follows that the set

$$\{\mathbf{x} \in \mathbb{R}^d \mid -\mathbf{W}\mathbf{x} \geq \mathbf{1}, \mathbf{A}\mathbf{x} \geq \mathbf{0}\} \subseteq \{\mathbf{x} \in \mathbb{R}^d \mid -\mathbf{W}\mathbf{x} > \mathbf{0}, \mathbf{A}\mathbf{x} \geq \mathbf{0}\}$$

is empty and hence the following linear program does not admit a feasible solution.

$$
\begin{aligned}
\min_{\mathbf{x}} \quad & 0 \\
\text{s.t.} \quad & \mathbf{A}\mathbf{x} \geq \mathbf{0} \\
& -\mathbf{W}\mathbf{x} \geq \mathbf{1}
\end{aligned}
$$

By strong duality, the dual linear program

$$
\begin{aligned}
\max_{\mathbf{y}} \quad & \mathbf{1}^T \mathbf{y} \\
\text{s.t.} \quad & \mathbf{z}^T \mathbf{A} - \mathbf{y}^T \mathbf{W} = \mathbf{0} \\
& \mathbf{y}, \mathbf{z} \geq \mathbf{0}
\end{aligned}
$$

has either no feasible solution or its objective value is unbounded. Since $\mathbf{y} = \mathbf{0}$ and $\mathbf{z} = \mathbf{0}$ yields a feasible solution, the latter is the case. In particular, there is a ray $\rho$ of the cone $\{(\mathbf{y}, \mathbf{z}) \in \mathbb{R}^{n+m} \mid \mathbf{z}^T \mathbf{A} - \mathbf{y}^T \mathbf{W} = \mathbf{0}, \mathbf{y}, \mathbf{z} \geq \mathbf{0}\}$ such that the objective value is unbounded on $\rho$. The dimension of the subspace $\{(\mathbf{y}, \mathbf{z}) \in \mathbb{R}^{n+m} \mid \mathbf{z}^T \mathbf{A} - \mathbf{y}^T \mathbf{W} = \mathbf{0}\}$ is at least $n + m - d$. Therefore, $\rho$ as a 1-dimensional subspace lies in the intersection of at least $n + m - d - 1$ many hyperplanes of the form $\{(\mathbf{y}, \mathbf{z}) \in \mathbb{R}^{n+m} \mid (\mathbf{y}, \mathbf{z})_i = 0\}$ and hence in the intersection of at least $n - d - 1$ many hyperplanes of the form $\{(\mathbf{y}, \mathbf{z}) \in \mathbb{R}^{n+m} \mid \mathbf{y}_i = 0\}$. Let $B \subseteq [n]$ be the set of size at least $n - d - 1$ such that $\rho \subseteq \bigcap_{i \in B} \{(\mathbf{y}, \mathbf{z}) \in \mathbb{R}^{n+m} \mid \mathbf{y}_i = 0\}$ (this can be computed in polynomial time; see, e.g., (Schrijver, 1986, Corollary 14.1g)) and $A := [n] \setminus B$ its complement.

It follows that $|A| \leq d + 1$ and that the objective value of the following LP is still unbounded.

$$
\begin{aligned}
\max_{\mathbf{y}} \quad & \mathbf{1}^T \mathbf{y} \\
\text{s.t.} \quad & \mathbf{z}^T \mathbf{A} - \mathbf{y}_A^T \mathbf{W}_A = \mathbf{0} \\
& \mathbf{y}, \mathbf{z} \geq \mathbf{0}
\end{aligned}
$$

Again, by strong duality, this implies that the set

$$S := \{\mathbf{x} \in \mathbb{R}^d \mid -\mathbf{W}_A \mathbf{x} \geq \mathbf{1}, \mathbf{A}\mathbf{x} \geq \mathbf{0}\}$$

is empty. Assume that the cone $\{\mathbf{x} \in \mathbb{R}^d \mid -\mathbf{W}_A \mathbf{x} > \mathbf{0}, \mathbf{A}\mathbf{x} \geq \mathbf{0}\}$ contains an element $\mathbf{x}$. Let $k := \min_{i \in [d]}\{(-\mathbf{W}_A \mathbf{x})_i\}$. Then $\frac{1}{k} \cdot \mathbf{x} \in S$, which is a contradiction. Hence, the set

$$\{\mathbf{x} \in \mathbb{R}^d \mid -\mathbf{W}_A \mathbf{x} > \mathbf{0}, \mathbf{A}\mathbf{x} \geq \mathbf{0}\}$$

is empty, which means that $C \subseteq \bigcup_{i \in A} H_{\mathbf{w}_i}^+$, proving the claim. $\qquad\square$

#### A.2.2   PROOF OF THEOREM 10

**Lemma 20.** *Algorithm 1 is correct.*

*Proof.* Assume that the algorithm outputs some $\mathbf{x} \in \mathbb{R}^d$. If $\mathbf{x}$ was returned in Line 1 or Line 2, then this is clearly correct since either no cell has rank $d$ or there is a rank-0 cell. If $\mathbf{x}$ was returned by some call of FindCell (Algorithm 2) in Line 4, then this is correct since $\mathrm{rank}(W_{\mathbf{x}}) \leq \mathrm{rank}(C) < d$, where $W_{\mathbf{x}} := \{\mathbf{w}_i \in W \mid \mathbf{w}_i^T \mathbf{x} \geq 0\}$. The second inequality holds since otherwise the algorithm would have returned "No" in Line 1. For the first inequality, we first observe the invariant that $W \setminus \mathrm{span}(C) \subseteq M \subseteq W$ holds at any time during execution of FindCell. This is clear for the initial call of FindCell in Line 3 in Algorithm 1. Also, within FindCell the property holds after Line 3 and for the recursive calls in Line 7. Now, $\mathrm{rank}(W_{\mathbf{x}}) \leq \mathrm{rank}(C)$ holds since $\mathbf{m}_i^T \mathbf{x} < 0$ for all $i \in [m]$ implies that $W_{\mathbf{x}} \cap M = \emptyset$, and thus, by our invariant, we have $C \subseteq W_{\mathbf{x}} \subseteq \mathrm{span}(C)$.

For the opposite direction, assume that there is an $\mathbf{x} \in \mathbb{R}^d$ such that $k := \mathrm{rank}(W_{\mathbf{x}}) < d$. If $k = 0$, then Algorithm 1 correctly returns some point of a rank-0 cell in Line 2. If $k > 0$, then $W_{\mathbf{x}}$ contains at least one vector $\mathbf{w}_i$. We claim that the call of FindCell in Line 3 will correctly return some point from a cell with rank at most $k$. To this end, we show that FindCell($C$, $M$) (Algorithm 2) returns a correct point whenever $C \subseteq W_{\mathbf{x}}$. Clearly, if $\mathrm{rank}(C) = k$, then $\mathbf{x}$ satisfies the conditions in Line 4 since $C \subseteq W_{\mathbf{x}}$, and thus $W_{\mathbf{x}} \subseteq \mathrm{span}(C)$ while $M \subseteq W \setminus \mathrm{span}(C)$ (due to Line 3 and the invariant on $M$). Hence, FindCell returns a correct point in this case.

Now consider the case $\mathrm{rank}(C) < k$. If some point is returned in Line 4, then this is correct (as already shown above). If no point satisfies the conditions in Line 4, then it holds $\{\mathbf{x} \mid \forall i \in [n] : \mathbf{c}_i^T \mathbf{x} \geq 0\} \subseteq \bigcup_{i \in I} H_{\mathbf{m}_i}^+$ and by Lemma 9, we can compute the set $A$ in Line 5. Note that $W_{\mathbf{x}} \cap \{\mathbf{m}_i \mid i \in A\} \neq \emptyset$. Hence, for at least one of the recursive calls in Line 7, it holds that $C \cup \{\mathbf{m}_i\} \subseteq W_{\mathbf{x}}$. Moreover, $\mathrm{rank}(C \cup \{\mathbf{m}_i\}) = \mathrm{rank}(C) + 1$ since $\mathbf{m}_i \notin \mathrm{span}(C)$ (due to Line 3). Hence, by induction, this call will return a correct point. $\square$

**Lemma 21.** *Algorithm 1 runs in $O((d+1)^d \cdot \mathrm{poly}(S))$ time, where $S$ denotes the input size.*

*Proof.* Let $S$ be the bit-length of $W$. Clearly, Line 1 and Line 2 can be done in $\mathrm{poly}(S)$ time via linear programming. As regards the running time of FindCell, note first that the recursion depth is at most $d$ since every recursive call increases the rank of $C$ (as already discussed in the proof of Lemma 21) and the recursion terminates when rank $d$ is reached. Moreover, each call of FindCell branches into at most $d + 1$ recursive calls, that is, the search tree has size at most $(d+1)^d$. Since all other computations within FindCell can be done in $\mathrm{poly}(S)$ time (using linear programming and Lemma 9), we obtain the desired running time. $\square$

## A.3 APPENDIX TO SECTION 5

### A.3.1 PROOF OF THEOREM 12

*Proof.* We reduce from the complement of 2-LAYER RELU POSITIVITY. Let $\mathbf{W}_1 \in \mathbb{R}^{n \times k}, \mathbf{W}_2 \in \mathbb{R}^{1 \times n}$ be an instance of 2-LAYER RELU POSITIVITY. Let the dimension $d$, the affine space $A$ (as a basis) and the point $\mathbf{z} \in \mathbb{R}^d$ be the output of the algorithm (on input $k$) that exists due to the fact that $(S_d)_{d \in \mathbb{N}}$ is a reasonable sequence of sets. More precisely, let $\varepsilon > 0$ be chosen such that

$$B = B_\varepsilon(\mathbf{z}) := \{\mathbf{x} \in A \mid \|\mathbf{x} - \mathbf{z}\|_2 < \varepsilon\} \subseteq S_d.$$

Let $P \colon \mathbb{R}^d \to A$ be the orthogonal projection to $A$ and $T \colon A \to \mathbb{R}^k$ an isometric isomorphism obtained by mapping the normalized basis of $A$ to the standard basis of $\mathbb{R}^k$. The composition $(T \circ P) \colon \mathbb{R}^d \to \mathbb{R}^k$ is an affine map and let it be given by a matrix $\mathbf{A} \in \mathbb{R}^{k \times d}$ and a vector $\mathbf{b} \in \mathbb{R}^d$. Then, $\widetilde{\mathbf{W}}_2 := \mathbf{W}_2, \widetilde{\mathbf{W}}_1 := \mathbf{W}_1 \cdot \mathbf{A}, \widetilde{\mathbf{b}}_1 := \mathbf{W}_1 \cdot \mathbf{b} - \mathbf{W}_1 \cdot \mathbf{A} \cdot \mathbf{z}, \widetilde{b}_2 := t$ form an instance of 2-LAYER RELU $(S,t)$-VERIFICATION and it holds that

$$\mathbf{W}_2 \cdot \phi_{\mathbf{W}_1}(\mathbf{A}\mathbf{x} + \mathbf{b}) > 0 \iff \widetilde{\mathbf{W}}_2 \cdot \phi_{\widetilde{\mathbf{W}}_1, \widetilde{\mathbf{b}}_1}(\mathbf{z} + \mathbf{x}) + \widetilde{b}_2 > t. \tag{1}$$

For the correctness of the reduction, assume that there is a $\mathbf{y} \in \mathbb{R}^k$ such that $\mathbf{W}_2 \cdot \phi_{\mathbf{W}_1}(\mathbf{y}) > 0$. Then, by the positive homogeneity of the map $\mathbf{x} \mapsto \mathbf{W}_2 \cdot \phi_{\mathbf{W}_1}(\mathbf{x})$, we also have that $\mathbf{W}_2 \cdot \phi_{\mathbf{W}_1}(\mathbf{y}') > 0$ for $\mathbf{y}' := \frac{\varepsilon}{2} \frac{\mathbf{y}}{\|\mathbf{y}\|}$. Since $T$ is an isometric isomorphism, there is an $\mathbf{x} \in A$ with $\|\mathbf{x}\| = \frac{\varepsilon}{2}$ such that $T(\mathbf{x}) = \mathbf{y}'$. Hence, $\mathbf{z} + \mathbf{x} \in B \subseteq S_d$ and Equation (1) implies that $\widetilde{\mathbf{W}}_2 \cdot \phi_{\widetilde{\mathbf{W}}_1, \widetilde{\mathbf{b}}_1}(\mathbf{z} + \mathbf{x}) + \widetilde{b}_2 > t$.

Conversely, if there is an $\mathbf{x} \in S_d$ such that $\widetilde{\mathbf{W}}_2 \cdot \phi_{\widetilde{\mathbf{W}}_1, \widetilde{\mathbf{b}}_1}(\mathbf{x}) + \widetilde{b}_2 > t$, then Equation (1) implies that $\mathbf{W}_2 \cdot \phi_{\mathbf{W}_1}(\mathbf{A}(\mathbf{x} - \mathbf{z}) + \mathbf{b}) > 0$, concluding the proof. $\square$

### A.3.2 Proof of Lemma 14

*Proof.* a) If there is a $\mathbf{v}^+ \in \mathbb{R}^d$ with $f_0(\mathbf{v}^+) = a > 0$, then for all $b \in [0, \infty)$ it holds that $f_0(\frac{b}{a}\mathbf{v}^+) = b$ due to positive homogeneity of $f_0$ (analogously for all $b \in (-\infty, 0]$ with $\mathbf{v}^-$). The other direction is trivial.

b) We start with some preliminary observations. First, there is a constant $C \in \mathbb{R}$ that only depends on the weights and biases of $f$ such that $\|f - f_0\|_\infty \leq C$ (Hertrich et al. (2021) Proposition 2.3). Moreover, due to continuity of $f$ it holds that $f$ is surjective if and only if for every $a < b \in \mathbb{R}$ there are $a', b' \in \mathbb{R}$ with $a' < a$ and $b' > b$ such that $a', b' \in f(\mathbb{R}^d)$.

Now, for the first direction, assume that $f$ is surjective. Then there are $a < -C$ and $b > C$ such that $a, b \in f(\mathbb{R}^d)$. Since $\|f - f_0\|_\infty \leq C$, it follows that there are $\mathbf{v}^+$ and $\mathbf{v}^-$ with $\phi_0(\mathbf{v}^+) > 0$ and $\phi_0(\mathbf{v}^-) < 0$, implying with a) that $\phi_0$ is surjective.

For the converse direction, let $a < 0 < b$. By surjectivity of $f_0$, we have that $a - 2C \in f_0(\mathbb{R}^d)$ and $b + 2C \in f_0(\mathbb{R}^d)$. Hence, since $\|f - f_0\|_\infty \leq C$, it follows that there are $a' < a$ and $b' > b$ such that $a', b' \in f(\mathbb{R}^d)$, implying surjectivity of $f$.

c) Follows directly from a) and Observation 3. $\qquad\square$

### A.3.3 Proof of Lemma 16

*Proof.* We define the sets $I^+ := \{i \in [n] \mid (\mathbf{W}_2)_i = 1\}$ and $I^- := [n] \setminus I^+$ and let $\mathbf{w}_1, \ldots, \mathbf{w}_n$ be the rows of $\mathbf{W}_1$.

First, we can assume that for any $\mathbf{v} \in \mathbb{R}^d$ there is at most one $i \in [n]$ such that $\mathbf{w}_i \in \mathrm{pos}(\mathbf{v}) := \{\lambda\mathbf{v} \mid \lambda \geq 0\}$. To see this, let $\mathbf{w}_i \in \mathrm{pos}(\mathbf{w}_j)$ for some $i, j \in [n]$. If $i, j \in I^+$, then we can simply delete the rows $\mathbf{w}_i$ and $\mathbf{w}_j$ and add a new row $\mathbf{w}_i + \mathbf{w}_j$ without changing the map $f$ (clearly the same works for $i, j \in I^-$). If $i \in I^+$ and $j \in I^-$, then we can delete the rows $\mathbf{w}_i$ and $\mathbf{w}_j$ and, if $\|\mathbf{w}_j\|_2 \leq \|\mathbf{w}_i\|_2$, add a new row $\mathbf{w}_i - \mathbf{w}_j$ with output weight 1 or add a new row $\mathbf{w}_j - \mathbf{w}_i$ with output weight $-1$ if $\|\mathbf{w}_i\|_2 \leq \|\mathbf{w}_j\|_2$ without changing the map $f$. Note that we can transform any matrix $\mathbf{W}_1$ to such a form in polynomial time.

Now, if for every row $\mathbf{w}_j$ there is a row $\mathbf{w}_{j'}$ such that $\mathbf{w}_j = -\mathbf{w}_{j'}$ and $(\mathbf{W}_2)_j = -(\mathbf{W}_2)_{j'}$, it follows that $f$ is a linear map and hence we can easily check whether it is the zero map. Otherwise, assume that for a row $\mathbf{w}_j$ there is no such row $\mathbf{w}_{j'}$. Then $\mathbf{w}_j$ induces a hyperplane $H_j := \{\mathbf{x} \in \mathbb{R}^d \mid \mathbf{w}_j\mathbf{x} = 0\}$ such that $\phi$ is not linear in every open neighborhood of any $\mathbf{x} \in H_j$ and hence the map $f$ is not linear and in particular cannot be the zero map. Thus, we can check in polynomial time whether $f = 0$.

Now in the case of $f \neq 0$, let $I_j := \{i \in [n] \mid \mathbf{w}_i \in \mathrm{span}(\mathbf{w}_j)\}$ and note that $|I_j| \leq 2$. For $i \in [n]$, we define the hyperplane $H_i := \{\mathbf{x} \in \mathbb{R}^d \mid \mathbf{w}_i\mathbf{x} = 0\}$. By definition, there exists an $\mathbf{x} \in H_j \setminus \left(\bigcup_{i \in [n] \setminus I_j} H_i\right)$. Now, for

$$\varepsilon := \min\{1, {}^1\!/_2 \min_{i \in [n] \setminus I_j} \min_{\mathbf{y} \in H_i} \|\mathbf{x} - \mathbf{y}\|_2 > 0\}$$

it holds that $\{\mathbf{x} + \delta\mathbf{w}_j, \mathbf{x} - \delta\mathbf{w}_j\} \subset \mathbb{R}^d \setminus \left(\bigcup_{i \in [n] \setminus I_j} H_i\right)$ for all $\delta \in (0, \varepsilon)$. Let $\mathbf{x}' := \mathbf{x} + \varepsilon\mathbf{w}_j$ and $\mathbf{x}'' := \mathbf{x} - \varepsilon\mathbf{w}_j$ and let $I' := \{i \in [n] \mid \mathbf{w}_i\mathbf{x}' > 0\}$ and $I'' := \{i \in [n] \mid \mathbf{w}_i\mathbf{x}'' > 0\}$.

We will argue now that either in the cell $C' \in \Sigma_{\mathbf{W}_1}$ containing $\mathbf{x}'$ or in the cell $C'' \in \Sigma_{\mathbf{W}_1}$ containing $\mathbf{x}''$ we find the desired $\mathbf{x}^*$ with $f(\mathbf{x}^*) \neq 0$. Note that it is sufficient to prove that $f$ cannot be the zero map on $C'$ and $C''$. We prove this by showing that

$$\mathbf{W}_2 \circ (\mathbf{W}_1)_{C'} - \mathbf{W}_2 \circ (\mathbf{W}_1)_{C''} \neq \mathbf{0}.$$

Note that $(I' \cup I_j) \setminus \{j\} = I''$. If $I_j = \{j\}$, then

$$\left(\sum_{i \in I' \cap I^+} \mathbf{w}_i - \sum_{i \in I' \cap I^-} \mathbf{w}_i\right) - \left(\sum_{i \in I'' \cap I^+} \mathbf{w}_i - \sum_{i \in I'' \cap I^-} \mathbf{w}_i\right) = \pm\mathbf{w}_j \neq \mathbf{0}.$$

If $I_j = \{j, j'\}$, then we have $\mathbf{w}_{j'} = -\lambda \mathbf{w}_j$ for some $\lambda > 0$ and hence

$$\left( \sum_{i \in I' \cap I^+} \mathbf{w}_i - \sum_{i \in I' \cap I^-} \mathbf{w}_i \right) - \left( \sum_{i \in I'' \cap I^+} \mathbf{w}_i - \sum_{i \in I'' \cap I^-} \mathbf{w}_i \right) = \pm \mathbf{w}_j \pm \mathbf{w}_{j'}$$

equals $\mathbf{0}$ if and only if $\lambda = 1$ and $(\mathbf{W}_2)_j = -(\mathbf{W}_2)_{j'}$, which we assumed not to be the case. $\qquad \square$

### A.3.4   PROPOSITION 22

In order to prove that POSITIVE CUT is NP-complete, we reduce from the following NP-hard problem (Bonsma et al. (2010)).

    DENSEST CUT

    **Input:**    A graph $G = (V, E)$ and $t \in \mathbb{Q} \cap [0, 1]$.

    **Question:**  Is there a subset $S \subseteq V$ such that $\frac{|E(S, V \setminus S)|}{|S| \cdot |V \setminus S|} > t$?

**Proposition 22.** POSITIVE CUT *is NP-complete.*

*Proof.* We reduce from DENSEST CUT. Let $(V, E)$ be a graph and $t = \frac{a}{b}$ with $a < b \in \mathbb{N}$ and let $w(F) := \sum_{e \in F} w(e)$. We construct the complete graph $K_{|V|} = (V, E' := \binom{V}{2})$ with edge weights $w \colon E' \to \mathbb{Z}$ by

$$w(\{i, j\}) = \begin{cases} -ab & \{i, j\} \notin E \\ (b - a)b & \{i, j\} \in E \end{cases}.$$

Note that for any $S \subseteq V$, it holds that

$$w(E'(S, V \setminus S)) = (b - a)b \cdot |E(S, V \setminus S)| - ab(|S| \cdot |V \setminus S| - |E(S, V \setminus S)|)$$
$$= b^2 \cdot |E(S, V \setminus S)| - ab \cdot |S| \cdot |V \setminus S|.$$

Hence, we have

$$w(E'(S, V \setminus S)) > 0 \iff \frac{b^2 \cdot |E(S, V \setminus S)|}{ab \cdot |S| \cdot |V \setminus S|} > 1 \iff \frac{|E(S, V \setminus S)|}{|S| \cdot |V \setminus S|} > \frac{a}{b} = t,$$

proving the correctness of the reduction. $\qquad \square$

### A.3.5   PROOF OF THEOREM 11

*Proof.* The problem is contained in NP since, by Lemma 14, it is sufficient to define a ray $\rho$ as a certificate for positivity. Since rays are one-dimensional subspaces, they are the intersection of $d - 1$ hyperplanes corresponding to rows of $\mathbf{W}_1$. Hence, the $2(d - 1)$ rows of $\mathbf{W}_1$ that determine $\rho$ form a polynomial-time verifiable certificate.

For the NP-hardness, we reduce from POSITIVE CUT. Given a weighted graph $(G = (V, E), w)$ with $V = [d]$ and $|E| = n$, we define the matrices $\mathbf{W}_1 \in \mathbb{R}^{2n \times d}$ and $\mathbf{W}_2 \in \mathbb{R}^{1 \times 2n}$ as follows: For each $e = \{i, j\} \in E$, $\mathbf{W}_1$ contains two rows $\mathbf{w}_e$ and $\mathbf{w}'_e$, where

$$(\mathbf{w}_e)_\ell := \begin{cases} 1, & \ell = i \\ -1, & \ell = j \\ 0, & \text{else} \end{cases} \text{ and } \mathbf{w}'_e := -\mathbf{w}_e.$$

The corresponding entries of $\mathbf{W}_2$ are set to $w(e)$. Thus, the 2-layer ReLU neural network computes the map $f \colon \mathbb{R}^d \to \mathbb{R}$ with

$$f(\mathbf{x}) = \sum_{\{i,j\} \in E} w(\{i, j\}) \cdot ([\mathbf{x}_i - \mathbf{x}_j]_+ + [\mathbf{x}_j - \mathbf{x}_i]_+).$$

For the correctness, we start with some preliminary observations. For a subset $S \subseteq V$, let $\mathbf{r}_S := \sum_{i \in S} \mathbf{e}_i \in \mathbb{R}^d$ and $\mathbf{r}'_S := -\sum_{i \in S \setminus V} \mathbf{e}_i \in \mathbb{R}^d$ and note that

$$[(\mathbf{r}_S)_i - (\mathbf{r}_S)_j]_+ + [(\mathbf{r}_S)_j - (\mathbf{r}_S)_i]_+ = \begin{cases} 0, & \{i, j\} \notin E(S, V \setminus S) \\ 1, & \{i, j\} \in E(S, V \setminus S) \end{cases}$$

as well as

$$[(\mathbf{r}'_S)_i - (\mathbf{r}'_S)_j]_+ + [(\mathbf{r}'_S)_j - (\mathbf{r}'_S)_i]_+ = \begin{cases} 0, & \{i,j\} \notin E(S, V \setminus S) \\ 1, & \{i,j\} \in E(S, V \setminus S) \end{cases}$$

and therefore

$$f(\mathbf{r}_S) = f(\mathbf{r}'_S) = \sum_{\{i,j\} \in E(S, V \setminus S)} w(\{i,j\}) = w(E(S, V \setminus S)).$$

As regards the correctness, if $(G, w)$ is a yes-instance, that is, there exists a subset $S \subseteq V$ with $w(E(S, V \setminus S)) > 0$, then $f(\mathbf{r}_S) > 0$.

Conversely, assume that there is a $\mathbf{v} \in \mathbb{R}^d$ with $f(\mathbf{v}) > 0$. Let $\pi \in \mathcal{S}_{d+1}$ be a permutation such that $\mathbf{v}_{\pi(1)} \le \mathbf{v}_{\pi(2)} \le \cdots \le \mathbf{v}_{\pi(d+1)}$, where $\mathbf{v}_{d+1} := 0$. Since all breakpoints $\mathbf{x}$ of $f_{\mathbf{W}_1}$ satisfy $\mathbf{x}_i = \mathbf{x}_j$ for some $i$ and $j$, the map $f_{\mathbf{W}_1}$ is linear within the pointed $d$-dimensional cone

$$C := \{\mathbf{x} \in \mathbb{R}^d \mid \mathbf{x}_{\pi(1)} \le \cdots \le \mathbf{x}_{\pi(d+1)}\} = \bigcap_{i=1}^{d} \{\mathbf{x} \in \mathbb{R}^n \mid \mathbf{x}_{\pi(i)} \le \mathbf{x}_{\pi(i+1)}\},$$

where again $\mathbf{x}_{d+1} := 0$. Hence, by linearity of the output layer and Observation 3, the value $\phi(\mathbf{v})$ is a conical combination of the values of $f$ on the ray generators of the rays of $C$ and hence there is ray generator $\mathbf{r}$ of $C$ such that $f(\mathbf{r}) > 0$. All rays of $C$ are the intersection of $C$ and $n - 1$ hyperplanes of the form $\{\mathbf{x} \in \mathbb{R}^d \mid \mathbf{x}_{\pi(i)} = \mathbf{x}_{\pi(i+1)}\}$. We denote these rays by

$$\rho_k := \{\mathbf{x} \in \mathbb{R}^d \mid \mathbf{x}_{\pi(1)} = \cdots = \mathbf{x}_{\pi(k-1)} \le \mathbf{x}_{\pi(k)} = \cdots = \mathbf{x}_{\pi(d+1)}\}$$

for $k \in [d]$. Let $k \in [d]$ and $S = \{\pi(k), \ldots, \pi(d)\}$. If $\pi^{-1}(d+1) < k$, then $\mathbf{r}_S$ generates $\rho_k$ and otherwise $\mathbf{r}'_S$ generates $\rho_k$. Since $\phi$ is positive on one of these ray generators, we can conclude that there is a $S \subseteq V$ such that $f(\mathbf{r}_S) = f(\mathbf{r}'_S) > 0$ which implies $w(E(S, V \setminus S)) > 0$. $\qquad\square$

