# OpenReview forum: "Complexity of Injectivity and Verification of ReLU Neural Networks"
_ICLR.cc/2025/Conference — Submitted to ICLR 2025_

### Official Review · Reviewer_A44m · 2024-10-18

**Soundness:** 2
**Presentation:** 1
**Contribution:** 2
**Rating:** 5
**Confidence:** 3

**Summary:**

This paper studies the complexity class of determining injectivity, surjectivity and robustness verification of ReLU networks. It proves the corresponding complexity class of each problem.

**Strengths:**

This paper, if the proofs are correct, should be quite novel and point out the algorithmic complexity of some problems. However, due to the writing, it is impossible for me to check the proofs end to end, even as a researcher who authored many purely theoretical papers.

**Weaknesses:**

The first thing bothers me is the problem itself: determining the injectivity of a ReLU network. First, when the network maps 1d input to 1d output, injectivity and continuity (naturally hold for ReLU networks) implies monotonicity. Second, in high dimensions, a function mapping from $\mathbb{R}^m$ to $\mathbb{R}^d$ with $m>d$ cannot be injective. Third, if we only consider injectivity inside a compact input set, as long as at least one neuron could be dead in the network inside this input set, this ReLU layer is not injective, and the composition of the ReLU layer with any other functions cannot be injective. Based on this facts, I fail to see how the conclusion could be correct: deciding the injectivity of a mapping from $\mathbb{R}^m$ to $\mathbb{R}$ is hard. Note that deciding whether all neurons cannot be dead in a ReLU network is easy: if all neurons in previous layers are active, they are then linear, and can be collectively merged together into a single linear layer; then, applying IBP [1] on a single linear layer results in precise bounds [8], and we can easily derive whether neurons in the current considered ReLU layer are all active.

Second, the authors write about robustness verification, but other than citing Katz et al., most of the related works are nonsense. For example, Line 110 mentions that heuristic approaches for verification exist, but instead of citing many established verification algorithms, they cite some random papers that aren't important enough. For example, they should at least cite methods such as IBP [1], DeepPoly [2], DeepZ [3], general cutting planes [4], multi-neuron certification [5], etc. and libraries such as LiRPA [6] and CTBench [7]. Line 115 also cites some random papers to support that the expressivity of ReLU nets is important.

Third, the authors state that the conclusion of Katz et al. only applies to polyhedra input sets and does not generalize, which I kindly disagree with.  Just like the authors did to generalize, a well-behaved input set contains lots of polyhedra, and Katz et al. have proven that even verification in those subsets is hard. Therefore, I fail to see how their conclusions generalize the result from Katz et al.

**Reference**

[1] Sven Gowal, Krishnamurthy Dvijotham, Robert Stanforth, Rudy Bunel, Chongli Qin, Jonathan Uesato, Relja Arandjelovic, Timothy A. Mann, and Pushmeet Kohli. On the effectiveness of interval bound propagation for training verifiably robust models.

[2] Gagandeep Singh, Timon Gehr, Markus Püschel, and Martin Vechev. An abstract domain for
certifying neural networks.

[3] Eric Wong, Frank R. Schmidt, Jan Hendrik Metzen, and J. Zico Kolter. Scaling provable adversarial defenses.

[4] Huan Zhang, Shiqi Wang, Kaidi Xu, Linyi Li, Bo Li, Suman Jana, Cho-Jui Hsieh, and J. Zico Kolter. General cutting planes for bound-propagation-based neural network verification.

[5] Claudio Ferrari, Mark Niklas Müller, Nikola Jovanovic, and Martin T. Vechev. Complete verification via multi-neuron relaxation guided branch-and-bound.

[6] Kaidi Xu, Zhouxing Shi, Huan Zhang, Yihan Wang, Kai-Wei Chang, Minlie Huang, Bhavya Kailkhura, Xue Lin, and Cho-Jui Hsieh. Automatic perturbation analysis for scalable certified robustness and beyond.

[7] Yuhao Mao, Stefan Balauca, Martin Vechev. CTBENCH: A Library and Benchmark for Certified Training.

[8] Yuhao Mao, Mark Niklas Mueller, Marc Fischer, and Martin Vechev. Understanding certified training with interval bound propagation.

**Questions:**

See weakness. Overall, I suggest the authors at least improve the writing of their proofs, e.g., making them more intuitive and highlighting the intuition and main results instead of throwing everything at the reader. This brings bad practice to the theory field.

---

> ### Author Response · Authors · 2024-11-22
>
> Thank you for providing helpful feedback that helps us to improve our paper.
>
> *The first thing bothers me is the problem itself: determining the injectivity of a ReLU network. First, when the network maps 1d input to 1d output, injectivity and continuity (naturally hold for ReLU networks) implies monotonicity.*
>
> This is correct and was already known before, since the problem is polynomial-time solvable for every constant input dimension. Note that an easy special case does not mean that the problem is not hard.
>
> *Second, in high dimensions, a function mapping from $\mathbb{R}^m$
> to $\mathbb{R}^d$ with $d > m$ cannot be injective.*
>
> This is also correct, but does not say anything about the case $d \leq m$ and in particular not, whether the problem is hard or not. Again, an NP-hard problem can have easy special cases.
>
> *Third, if we only consider injectivity inside a compact input set, as long as at least one neuron could be dead in the network inside this input set, this ReLU layer is not injective, and the composition of the ReLU layer with any other functions cannot be injective. Based on this facts, I fail to see how the conclusion could be correct: deciding the injectivity of a mapping from to $\mathbb{R}^m$
> to $\mathbb{R}$ is hard. Note that deciding whether all neurons cannot be dead in a ReLU network is easy: if all neurons in previous layers are active, they are then linear, and can be collectively merged together into a single linear layer; then, applying IBP [1] on a single linear layer results in precise bounds [8], and we can easily derive whether neurons in the current considered ReLU layer are all active.*
>
> This is not correct. If a neuron is dead, the network can still be injective. As a simple example, consider the map from $\mathbb{R}$ to $\mathbb{R}^3$ given by $x \mapsto (\max(x,0),\max(-x,0),0)$. Note that the last neuron is dead everywhere but the map is still injective. Not every neuron necessarily has to compute an injective map when the function computed by the network is injective. The problem is indeed coNP-hard as our rigorous proof shows. Moreover, the computational complexity of deciding injectivity has been posed as an open question in the literature (Puthawala et al. 2022).
>
> *Second, the authors write about robustness verification, but other than citing Katz et al., most of the related works are nonsense. For example, Line 110 mentions that heuristic approaches for verification exist, but instead of citing many established verification algorithms, they cite some random papers that aren't important enough. For example, they should at least cite methods such as IBP [1], DeepPoly [2], DeepZ [3], general cutting planes [4], multi-neuron certification [5], etc. and libraries such as LiRPA [6] and CTBench [7]. Line 115 also cites some random papers to support that the expressivity of ReLU nets is important.*
>
> Thank you for mentioning those papers. We included them in the related work section in our revised version of the paper.
>
> *Third, the authors state that the conclusion of Katz et al. only applies to polyhedra input sets and does not generalize, which I kindly disagree with. Just like the authors did to generalize, a well-behaved input set contains lots of polyhedra, and Katz et al. have proven that even verification in those subsets is hard. Therefore, I fail to see how their conclusions generalize the result from Katz et al.*
>
> The reduction by Katz et al. specifically uses the unit cube as input set in order to encode the SAT problem. Their construction is based on a specific interplay between the input set and the network.
> For example, it is not clear what the network looks like when the input set is an arbitrary polyhedron or the unit L_2-ball.
> Therefore, the result of Katz et al. shows hardness only for cubes as part of the input. Our reduction, on the other hand, shows that verification is hard for basically every possible fixed input set (no matter which structure you assume). Thus, the hardness of verification is independent of the considered input set.
> In particular, our result implies hardness for all polyhedra and also all balls as input set, which is not covered by the result of Katz et al.
>
> *See weakness. Overall, I suggest the authors at least improve the writing of their proofs, e.g., making them more intuitive and highlighting the intuition and main results instead of throwing everything at the reader. This brings bad practice to the theory field.*
>
> We uploaded a revision of the paper, where we highlighted the main results, added further explanations to the proof sketches and included illustrations to provide more intuition.

---

> ### Comment · Reviewer_A44m · 2024-11-22
>
> Thanks for the reply. I am glad that the author clarified that the only interesting case is when input dimension is no larger than output dimension.
>
> I saw why the existence of dead neurons does not imply non-injectivity when input dimension is strictly smaller than output dimension. However, I have one additional question: if they are equal, does this imply non-injectivity?
>
> In addition, modern networks usually have a very large input dimension (e.g., 768 for MNIST) and a very small output dimension (e.g., 10 for MNIST). I don't see why deciding the injectivity is important for modern neural networks.
>
> I have raised my score to 3.

---

> ### Author Response · Authors · 2024-11-25
>
> We thank the reviewer for the answer.
>
> *I saw why the existence of dead neurons does not imply non-injectivity when input dimension is strictly smaller than output dimension. However, I have one additional question: if they are equal, does this imply non-injectivity?*
>
> A deep neural network with a dead neuron can be injective even if input and output dimensions are equal: Add an output layer to the example from our first response, which computes $\max(0,x)-\max(0,-x)+0$. This map is the identity and hence injective but still has a dead neuron. If there is only one hidden layer without a output layer, a dead neuron basically means projecting to a lower-dimensional space. Hence, in this case, if input and output dimensions are equal, it is trivially true that a dead neuron implies non-injectivity.
>
> *In addition, modern networks usually have a very large input dimension (e.g., 768 for MNIST) and a very small output dimension (e.g., 10 for MNIST). I don't see why deciding the injectivity is important for modern neural networks.*
>
> Injectivity is, of course, not important in classification tasks, where the goal is to map inputs to discrete categories and hence non-injectivity is desired.  However, in other areas of deep learning such as generative modeling, solving inverse problems, compressed sensing, and representation learning, injectivity is important because it ensures that distinct inputs are mapped to distinct outputs, preserving information and enabling tasks like invertibility, stable reconstruction, and interpretable latent spaces. In these contexts, injectivity of the map is essential for accurate and meaningful results. See the papers below for examples where invertibility is relevant. Moreover, we refer to the discussion of (Puthawala et al. 2022) about the relevance of injectivity in deep learning, where the computational complexity of deciding injectivity has been posed as an open question.
>
>
> - Patrick Putzky and Max Welling. 2019. Invert to learn to invert. Proceedings of the 33rd International Conference on Neural Information Processing Systems. Curran Associates Inc., Red Hook, NY, USA, Article 41, 446–456.
>
> - Durk P Kingma and Prafulla Dhariwal. Glow: Generative flow with invertible 1x1 convolutions. In S. Bengio, H. Wallach, H. Larochelle, K. Grauman, N. Cesa-Bianchi, and R. Garnett (eds.), Advances in Neural Information Processing Systems, volume 31. Curran Associates, Inc., 2018.
>
> - Lynton Ardizzone, Jakob Kruse, Carsten Rother, and Ullrich Kothe. Analyzing inverse problems
> with invertible neural networks. In International Conference on Learning Representations, 2019.
>
> - Ashish Bora, Ajil Jalal, Eric Price, Alexandros G. Dimakis, Compressed sensing using generative models. Proceedings of the 34th International Conference on Machine Learning, PMLR 70:537-546, 2017.
>
> - Michael Puthawala, Konik Kothari, Matti Lassas, Ivan Dokmanic, and Maarten V. de Hoop. Globally
> injective ReLU networks. Journal of Machine Learning Research, 23:105:1–105:55, 2022

---

> > ### Comment · Reviewer_A44m · 2024-11-26
> >
> > Thanks for the reply. This clarifies my concerns regarding the scope of this work.
> >
> > I decided to raise my score to 5 and not object to an accept if other reviewers feel it proper. However, this heavily reduced my confidence in this score, so I decreased my confidence to 3.

---

### Official Review · Reviewer_m4K3 · 2024-10-28

**Soundness:** 3
**Presentation:** 2
**Contribution:** 3
**Rating:** 6
**Confidence:** 3

**Summary:**

This paper states that the computational complexity of deciding infectivity for ReLU networks is coNP-complete. To prove this, they first show that $l$-layer ReLU injectivity is in coNP. They proceed then to show that it is also coNP-hard. However, they proceed to claim that for a single hidden layer, the problem is still tractable as long as the input dimensions are few.

They then study the network verification problem and prove that the network verification is coNP-hard for a general class of input domains. To do that, the authors define the notion of *reasonable* sequence of subsets and show that the the problem of deciding if a single hidden layer ReLU network outputting scalars is strictly positive is NP complete. They then proceed to prove coNP-hardness. The authors claim that the results highlight that the hardness of network verification is intrinsic to ReLU networks rather than the input domain.

**Strengths:**

The complexity of network analysis is an important area of research. In general, universal approximation and robustness are important results and an important theoretical justification in the field. The writing and presentation is mostly clear. While many results, especially in the beginning where attributed to prior works, later results appear mostly novel. Further, intuition was provided throughout the paper.

**Weaknesses:**

*A substantive assessment of the weaknesses of the paper. Focus on constructive and actionable insights on how the work could improve towards its stated goals. Be specific, avoid generic remarks. For example, if you believe the contribution lacks novelty, provide references and an explanation as evidence; if you believe experiments are insufficient, explain why and exactly what is missing, etc.*

Regarding the presentation, i encourage the authors to lighten the paper with some illustrations where appropriate. Further, it would be good, to have some intuition regarding the reduction to the graph problems in the paper. However, the biggest weakness of the paper is the missing related work. It is unclear how it relates to results on universal approximation for networks certified with convex relaxations like intervals [1,2,3,4] and specifically the hardness result presented in [2].

Further, the claims made in the conclusion appear too strong to me (see questions).

Missing related work:
1. Baader et. al. "[Universal Approximation with Certified Networks](https://openreview.net/forum?id=B1gX8kBtPr)", ICLR 2020
2. Zi et al. "[Interval universal approximation for neural networks](https://dl.acm.org/doi/abs/10.1145/3498675)", POPL 2022
3. Mirman et al. "[The Fundamental Limits of Neural Networks for Interval Certified Robustness](https://openreview.net/forum?id=fsacLLU35V)", TMLR 2022
4. Baader et. al. "[Expressivity of ReLU-Networks under Convex Relaxations](https://openreview.net/forum?id=awHTL3Hpto)", ICLR 2024

**Questions:**

- The claim in the conclusion to exclude polynomial-time algorithms seems to be too strong based on the analysis presented, as worst case complexity was analyzed. How does this claim compare to the results of [1,2,3,4]?
- Do the authors have an intuition how their bound relate to certification in practice? I.e. not assuming worst case networks or assuming that it is fine that certification sometimes failes / times out?
- Can the authors provide some intuition regarding the reduction to the graph problems in the main paper?

**Minor questions**
- For Definition 2, is there a reason why you decided not to write $\phi_{W_l, b_l} \circ \cdots \circ \phi_{W_1, b_1}$ and instead spell $\phi_{W_l, b_l}$ out?
- In Line 440 pint a) - i suppose it should be $v^+, v^-$, yes?

---

> ### Author Response · Authors · 2024-11-22
>
> Thank you for providing helpful feedback that helps us to improve our paper.
>
> *Regarding the presentation, i encourage the authors to lighten the paper with some illustrations where appropriate. Further, it would be good, to have some intuition regarding the reduction to the graph problems in the paper.*
>
> We included illustrations of the geometric definitions related to ReLU neural networks and of the reduction from the Acyclic 2-Disconnection problem to ReLU-Layer Non-Injectivity. We also added further explanation to the proof sketches making them more intuitive.
>
> *However, the biggest weakness of the paper is the missing related work. It is unclear how it relates to results on universal approximation for networks certified with convex relaxations like intervals [1,2,3,4] and specifically the hardness result presented in [2].*
>
>
> We extended our discussion of related work (see also answers to your questions below).
> The hardness result for the range approximation problem in [2] is not comparable to our result since it holds for networks with three layers with three different very specific activation functions which are not ReLU activation.
>
> *The claim in the conclusion to exclude polynomial-time algorithms seems to be too strong based on the analysis presented, as worst case complexity was analyzed. How does this claim compare to the results of [1,2,3,4]?*
>
> Clearly, coNP-hardness only excludes polynomial time in the worst case. We mention this now in the conclusion. Our result does not exclude special cases of networks that can be efficiently verified as described in the literature you mention. Indeed our result shows that one has to make (very) specific assumptions on the network in order to obtain tractable cases since it is not enough to only restrict the considered input set. Thus, our result confirms the approach to build special approximating networks that are efficiently verifiable. However, the hardness indicates that it might not be possible to construct such approximating networks in polynomial time.
>
> *Do the authors have an intuition how their bound relate to certification in practice? I.e. not assuming worst case networks or assuming that it is fine that certification sometimes failes / times out?*
>
> Our result shows that already simple networks (i.e. one hidden layer with sparse weights and one-dimensional output) can be the worst case. The practical implications, however, are not directly clear. It might still be unlikely to encounter these hard instances in practice. Also randomized approaches are not excluded.
>
> *Can the authors provide some intuition regarding the reduction to the graph problems in the main paper?*
>
> When reducing from the (di)graph problems, we encode the (di-)graph as a ReLU layer. Thereby, we introduce a neuron $\max \{x_i -x_j\} $ for every arc $(i,j) \in A$ (respectively two neurons for an edge).
>
> For the injectivity problem, the points whose coordinates are ordered with respect to an ordering $\pi$ form a cone $C_\pi$. In other words, the cone $C_\pi$ is defined by deciding on which side of the hyperplane $x_i -x_j=0$ the points are, i.e.,
> $C_\pi = \bigcap_{\pi(j) \leq \pi(i)} ${$x_i -x_j \geq 0$}. Hence, an ordering $\pi$ of the coordinates coorresponds to a subset of arcs $A_\pi =\{(i,j) \in A \mid \pi(i) \leq \pi(j) \}$ that are not active on $C_\pi$. The set $A_\pi$ is always acyclic by definition. We then show, that the digraph $(D, A \setminus A_\pi)$, i.e., removing the arcs that correspond to inactive neurons, is weakly connected if and only if the neural network is injective on the cone $C_\pi$. For an intuition for this last statement, we included an illustration for the 2-dimensional case. We also added some further explanation to the proof sketch.
>
> For the positivity problem, the hidden layer encodes the graph in the same way by replacing every edge with two directed arcs. The output weights of the neurons are the corresponding weights of the edges. We then identify a vectors $r_S$ for every subset $S \subseteq V$ such that the function value of the constructed ReLU network on $r_S$ is the cut value of the cut induced by $S$. Finally, we show that the function value of $x \in R^n$ is a conic combination of these cut values.
>
> See also the illustration that we added in the revision.
>
> *For Definition 2, is there a reason why you decided not to write $\phi_{W_l,b_l} \circ \phi_{W_1,b_1}$ and instead spell $\phi_{W_l,b_l}$ out?*
>
> Yes, the last layer is only an affine linear function without ReLU activation and hence we can not write it as $\phi_{W_l,b_l}$.
>
> *In Line 440 pint a) - i suppose it should be $v^+,v^-$, yes?*
>
> Yes.

---

> > ### Comment · Reviewer_m4K3 · 2024-12-01
> > **Rebuttal**
> >
> > I thank the authors for their clarifications and decided to raise my score.

---

### Official Review · Reviewer_XKa9 · 2024-10-31

**Soundness:** 3
**Presentation:** 2
**Contribution:** 3
**Rating:** 6
**Confidence:** 2

**Summary:**

This paper studies the complexity of deciding injectivity of a ReLU neural network. It is shown that the problem is coNP-complete. It is also shown that neural network verification (e.g., robustness verification) is coNP-hard, for "reasonable" sets in the input space (essentially, any set that contains a ball). Finally, it is also shown that deciding the surjectivity of a ReLU network is NP-complete. An explicit connection is drawn between deciding the existence of an input leading to a positive output under a scalar-valued 2-layer ReLU network, and the non-containment of two zonotopes. This gives rise to another explanation for the NP-hardness of determining positivity of a ReLU network.

**Strengths:**

The paper provides some answers to deep theoretical questions surrounding the complexity of certain decision problems in the analysis of ReLU neural networks. The results in this paper are certainly of interest to the ICLR community. Interesting open problems are posed to the community, in the conclusion of the paper.

**Weaknesses:**

I question the accessibility of this paper to the general ICLR audience. There is a lot of technical terminology used that is not defined for the general reader. The results in the main paper are all substantiated by rather imprecise proof sketches, rather than rigorous proofs. See also my Questions below.

**Questions:**

1. Line 79: "hardness results is" should be "hardness results are".
2. Line 137: You didn't define the acronym "wlog" (without loss of generality), which might confuse some readers that haven't seen it before.
3. Line 147: "forms a polyhedral fan" it would help the reader to recall the definition of a polyhedral fan.
4. Line 157: Do you mean to write $\phi\_{\mathbf{W}\_i,\mathbf{b}\_i}$ instead of $\phi\_{\mathbf{W}\_{n\_i},\mathbf{b}\_{n\_i}}$? Also, it seems like you should index your projection mappings as $\pi\_{i,j} \colon \mathbb{R}^{n\_i}\to \mathbb{R}$ (projection onto coordinate $j$), instead of $\pi\_{j} \colon \mathbb{R}^{n\_i}\to \mathbb{R}$, since its domain depends on $i$.
5. Line 168: The set $\mathcal{R}$ is technically not well-defined since $\mathbf{r}\_\rho$ is not unique, although I see what you're getting at. If you can easily update things to make $\mathcal{R}$ well-defined, that would be great. (For example, would it work to take $\mathbf{r}\_\rho$ as the unique unit vector generating the ray $\rho$?)
6. Observation 3: Again, you're using terminology that may not be immediately accessible/commonplace for most ICLR readers; you should define what an "extreme ray" of a polyhedral cone is.
7. Line 176: I think that this brief recap of P, NP, NP-complete, coNP-complete, XP, and FPT is great, but that it should be included earlier in the paper, before the terms XP and FPT are used (otherwise, you might lose some readers early on in the introduction as soon as XP and FPT are mentioned). For example, can shorten these descriptions even more, so that you can squeeze them into the introduction right before line 58?
8. Line 196: Again, I suggest you define your acronym LP (linear program).
9. Proposition 4: The provided proof sketch might cause confusion, since what you're saying looks like deciding non-injectivity is polynomial time as it reduces to a linear programming feasibility problem. So, why then would determining injectivity be in coNP based on this interpretation of your proof sketch?
10. Line 223: Again, you're using terminology that might be unfamiliar to some readers, without defining it (e.g., "digraph" instead of directed graph, weakly connected, acyclic, etc.---not all ICLR readers are graph theorists).
11. Line 242: Typo ("completenss" instead of "completeness").
12. Line 384: What do you mean by the notation $w(E(S,V \setminus S)) > 0$? Are you referring to the condition that $w(e) > 0$ for all edges $e \in E$ connecting a vertex $v\in V$ to a vertex $v' \in V\setminus S$? If so, it would be great to define this for the reader, since the notation seems sort of nonstandard.
13. As a follow-up to my above question, line 404 makes it look like $w(E(S,V\setminus S))$ equals the sum of weights of all edges connecting vertices in $V$ to vertices in $V\setminus S$. In my opinion, this makes it even like likely for the reader to assume what you mean by the notation $w(E(S, V\setminus S))$.
14. Line 440: Do you mean $\mathbf{v}^+,\mathbf{v}^{-}$ instead of $\mathbf{v}^+,\mathbf{v}^+$?

---

> ### Author Response · Authors · 2024-11-22
>
> Thank you for providing helpful feedback that helps us to improve our paper.
>
> *I question the accessibility of this paper to the general ICLR audience. There is a lot of technical terminology used that is not defined for the general reader. The results in the main paper are all substantiated by rather imprecise proof sketches, rather than rigorous proofs. See also my Questions below.*
>
> The revised version now contains definitions for all terminology used in the paper. It should now be fully self-contained.
>
> We also made the proof sketches a bit more precise and added illustrations in order to provide intuition. We also want to remark that for every statement, we give reference to a rigorous proof in the appendix. Due to the page limit, it is not possible to include all proofs in the main paper.
>
>
>
> *Line 79: "hardness results is" should be "hardness results are".*
>
> Done.
>
> *Line 137: You didn't define the acronym "wlog" (without loss of generality), which might confuse some readers that haven't seen it before.*
>
> Spelled it out.
>
> *Line 147: "forms a polyhedral fan" it would help the reader to recall the definition of a polyhedral fan.*
>
> Added definition.
>
> *Line 157: Do you mean to write $\phi_{W_i,b_i}$ instead $\phi_{W_{n_i},b_{n_i}}$ of ? Also, it seems like you should index your projection mappings as $\pi_{i,j} $(projection onto coordinate $j$), instead of $\pi_j$, since its domain depends on $i$.*
>
> It is correct that it depends on $i$, but in this case, we believe that it is more convenient to keep the notation short since there is no ambiguity.
>
> *Line 168: The set $\mathcal{R}$ is technically not well-defined since $r_\rho$ is not unique, although I see what you're getting at. If you can easily update things to make $\mathcal{R}$ well-defined, that would be great. (For example, would it work to take $\mathbf{r}_\rho$ as the unique unit vector generating the ray $\rho$?)*
>
> Adjusted it.
>
> *Observation 3: Again, you're using terminology that may not be immediately accessible/commonplace for most ICLR readers; you should define what an "extreme ray" of a polyhedral cone is.*
>
> Defined it.
>
> *Line 176: I think that this brief recap of P, NP, NP-complete, coNP-complete, XP, and FPT is great, but that it should be included earlier in the paper, before the terms XP and FPT are used (otherwise, you might lose some readers early on in the introduction as soon as XP and FPT are mentioned). For example, can shorten these descriptions even more, so that you can squeeze them into the introduction right before line 58?*
>
> We inserted brief explanations of FPT and XP in the introduction.
>
>
> *Line 196: Again, I suggest you define your acronym LP (linear program).*
>
> Spelled it out.
>
> *Proposition 4: The provided proof sketch might cause confusion, since what you're saying looks like deciding non-injectivity is polynomial time as it reduces to a linear programming feasibility problem. So, why then would determining injectivity be in coNP based on this interpretation of your proof sketch?*
>
> The proof sketch only argues that two polyhedra $P$ and $Q$ serve as a certificate for non-injectivity. This means that we can verify in polynomial time (by checking the feasibility of an LP) whether the map is not injective on $P$ and $Q$. Since we had to "guess" $P$ and $Q$ in the beginning, this does not yield a polynomial-time algorithm but proves that the problem is contained in coNP.
>
>
> *Line 223: Again, you're using terminology that might be unfamiliar to some readers, without defining it (e.g., "digraph" instead of directed graph, weakly connected, acyclic, etc.---not all ICLR readers are graph theorists).*
>
> We now use directed graph and defined weakly connected and acyclic.
>
> *Line 242: Typo ("completenss" instead of "completeness").*
>
> Done.
>
> *Line 384: What do you mean by the notation $w(E(S, V \setminus S)) > 0?$ Are you referring to the condition that $w(e) > 0$ for all edges $e \in E$ connecting a vertex $v \in V$ to a vertex $v' \in V \setminus S?$ If so, it would be great to define this for the reader, since the notation seems sort of nonstandard.
> As a follow-up to my above question, line 404 makes it look like $w(E(S, V \setminus S))$
> equals the sum of weights of all edges connecting vertices $V$ in to vertices in $V \setminus S$. In my opinion, this makes it even like likely for the reader to assume what you mean by the notation
> $w(E(S, V \setminus S))$.*
>
> By $E(S, V \setminus S)$ we meant the cut induced by $S$, i.e., the set of edges connecting vertices in $S$ with vertices in $V$ and $w(E(S, V \setminus S)) = \sum_{e \in E(S, V \setminus S)} w(e)$. We changed this in the paper and defined all the notation that we use.
>
> *Line 440: Do you mean $\mathbf{v}^+, \mathbf{v}^-$
> instead of $\mathbf{v}^+, \mathbf{v}^+$?*
>
> Yes, changed it.

---

> > ### Comment · Reviewer_XKa9 · 2024-11-22
> >
> > I thank the authors for their clarifications and edits to the paper. I have decided to increase my overall Rating by 1 point.

---

### Official Review · Reviewer_gZ12 · 2024-11-03

**Soundness:** 3
**Presentation:** 3
**Contribution:** 2
**Rating:** 3
**Confidence:** 3

**Summary:**

This paper investigates the computational challenges and theoretical limitations associated with key properties of ReLU neural networks: injectivity, verification, and surjectivity. These properties are important for applications that rely on the reversibility and reliability of neural networks, such as generative models, inverse problems, and safety-critical systems like autonomous vehicles. These findings deepen the understanding of ReLU network behavior, particularly for applications where robustness and accuracy are critical.

**Strengths:**

This paper makes a valuable contribution to the theoretical understanding of ReLU neural networks, particularly concerning injectivity, verification, and surjectivity, by rigorously analyzing these problems’ computational complexity. Below is an assessment based on originality, quality, clarity, and significance.

**1. Originality**: The paper tackles the novel and previously open question of the computational complexity of determining injectivity in ReLU networks, establishing its coNP-completeness. This result is particularly impactful because injectivity is a fundamental property for many applications where reversible or unique mapping is essential. The authors also creatively connect network surjectivity with a zonotope containment problem, revealing links between neural network analysis and computational geometry. Additionally, the paper’s formulation of the verification problem as a coNP-hard challenge within ReLU network structure itself—independent of specific input conditions—is an insightful addition. This treatment expands our understanding of why verification is so difficult across many practical applications, which sets this work apart from prior studies focused on domain-specific hardness assumptions.

**2. Quality**: The theoretical rigor of this work is apparent, with thorough proofs supporting the coNP-completeness of injectivity and the coNP-hardness of verification. By also providing parameterized complexity results for small-dimension cases, the authors have strengthened the practical relevance of their findings. The authors appropriately cite foundational work and carefully build upon it, ensuring that their proofs are grounded in established literature while providing new contributions. However, including more experimental or empirical validations of these theoretical results (e.g., demonstrating the parameterized algorithm on real or synthetic networks) could further reinforce the paper’s quality and applicability.

**3. Clarity**: The paper is generally well-structured and organized logically, progressing through definitions, results, proofs, and implications. The separation between theoretical complexity results and parameterized algorithms is well-done, allowing readers to follow the main contributions without confusion. However, the paper is dense, and certain sections, particularly around zonotope containment and parameterized complexity, may benefit from clearer explanations or examples to improve accessibility. For instance, more intuitive descriptions or visualizations of zonotope-related results could enhance understanding for readers who may not be familiar with computational geometry.

In summary, this paper is a substantial and rigorous contribution with theoretical insights that carry practical implications. Its originality, depth of analysis, and potential influence on further research in neural network properties make it a valuable addition to the field. Improvements in clarity and empirical validation would further enhance its impact, but overall, the paper is a high-quality work that addresses foundational questions in neural network theory.

**Weaknesses:**

This paper makes a significant theoretical contribution to the study of ReLU networks, but there are several areas where it could be further improved to enhance both its rigor and accessibility. Here are some specific areas for improvement:

**1. Limited Empirical Validation**: While the paper focuses on theoretical complexity results, an empirical component could improve its practical relevance. Implementing and testing the parameterized algorithm for injectivity on networks with small input dimensions or low neuron counts would provide insights into its effectiveness and limitations in real-world scenarios. Demonstrating the algorithm on a range of synthetic and small real datasets could validate the algorithm's runtime and parameter tractability claims, offering a concrete sense of its usability and performance. Additionally, practical experiments to showcase the difficulty of network verification and injectivity in deeper, multi-layer ReLU networks could help underscore the paper’s theoretical findings for readers in applied fields. An empirical analysis could reveal how these challenges scale and manifest in network training and verification processes.

**2. Limited Discussion on Practical Implications and Trade-offs**: While the paper addresses theoretical complexity boundaries, it lacks a nuanced discussion on the practical implications of these results for neural network practitioners. For example, the paper could delve into how the coNP-hardness of verification and injectivity should guide network architecture choices, especially in domains where reliability is paramount. It would be valuable to suggest actionable insights for practitioners, such as how to prioritize aspects like layer depth, dimensionality, or neuron count to balance network expressivity with computational feasibility. In the discussion of parameterized complexity results, additional analysis of where these results might realistically apply in common architectures could also be beneficial. For instance, clarifying the practical relevance of fixed-parameter tractability in applications where dimensionality is inherently high (e.g., vision tasks) would help readers understand where these findings might or might not be beneficial.

**3. Limited Exploration of Related Work in Neural Network Verification**: While the paper provides foundational complexity results, the treatment of related verification research could be expanded. There has been recent work on heuristics and approximation methods for ReLU network verification, which could provide important context here. Reviewing recent approaches to approximate verification or discussing alternative methods that offer partial or probabilistic verification could provide readers with additional avenues to explore beyond purely theoretical boundaries. Specifically, citing and discussing recent advancements in heuristic-based and constraint programming approaches to verification could frame the authors’ results within the current landscape of neural network verification tools and methods. This contextualization would help readers understand both the novelty and the limitations of purely theoretical approaches in applied contexts.

In summary, while the paper makes valuable theoretical contributions, it could benefit from empirical validation, improved accessibility of complex sections, and more thorough discussion of related work and practical implications. Addressing these weaknesses would broaden the impact and applicability of the findings, especially for audiences interested in practical neural network design and verification.

**Questions:**

1. The relationship between zonotope containment and ReLU network surjectivity is intriguing but could be more accessible. Could you provide a more intuitive explanation or visualization of this connection? Could you also provide illustrative examples of injective/surjective/bijective ReLU networks?

2. Given the coNP-hardness of injectivity and verification, how should practitioners approach network design to avoid infeasibility? Are there specific heuristics or guidelines that could help manage these complexities?

3. The paper focuses on theoretical hardness but doesn’t discuss how existing heuristic or approximate verification methods could be employed in light of your results [1-3]. Could you comment on where your results might influence the development or improvement of such methods?

4. Neural network verification is a computationally demanding process and has been shown to be NP-hard even for simple neural networks with a single hidden layer [1]. Could the authors clarify the novel contributions of your coNP-hardness result compared to existing NP-hardness results. Specifically, how your result provides new insights or implications for verification techniques that weren't apparent from previous hardness results?

5. Could you provide more insight into the chosen parameterization for injectivity (in terms of input dimension) and its realistic applicability? For example, how might this approach scale in contexts where input dimensionality is inherently high, such as in image recognition (e.g., MNIST images vs. high-resolution images)?

6. Given the exponential runtime for injectivity checking based on input dimension, is there potential for further optimization or approaches for improving the algorithm's efficiency? Additionally, could you discuss any possible conjecture about the lower bounds that might limit algorithmic improvement?

[1] Katz et al. 2017, Reluplex: An efficient SMT solver for verifying deep neural networks.
[2] Zhang et al. 2018, Efficient Neural Network Robustness Certification with General Activation Functions.
[3] Singh et al. 2018, Fast and Effective Robustness Certification.

---

> ### Author Response · Authors · 2024-11-22
> **1/2**
>
> Thank you for providing helpful feedback that helps us to improve our paper.
>
> *1. Limited Empirical Validation: While the paper focuses on theoretical complexity results, an empirical component could improve its practical relevance. Implementing and testing the parameterized algorithm for injectivity on networks with small input dimensions or low neuron counts would provide insights into its effectiveness and limitations in real-world scenarios. Demonstrating the algorithm on a range of synthetic and small real datasets could validate the algorithm's runtime and parameter tractability claims, offering a concrete sense of its usability and performance. Additionally, practical experiments to showcase the difficulty of network verification and injectivity in deeper, multi-layer ReLU networks could help underscore the paper’s theoretical findings for readers in applied fields. An empirical analysis could reveal how these challenges scale and manifest in network training and verification processes.*
>
> Experimental evaluation might be interesting but is clearly out of scope for this paper.
>
> *2. Limited Discussion on Practical Implications and Trade-offs: While the paper addresses theoretical complexity boundaries, it lacks a nuanced discussion on the practical implications of these results for neural network practitioners. For example, the paper could delve into how the coNP-hardness of verification and injectivity should guide network architecture choices, especially in domains where reliability is paramount. It would be valuable to suggest actionable insights for practitioners, such as how to prioritize aspects like layer depth, dimensionality, or neuron count to balance network expressivity with computational feasibility. In the discussion of parameterized complexity results, additional analysis of where these results might realistically apply in common architectures could also be beneficial. For instance, clarifying the practical relevance of fixed-parameter tractability in applications where dimensionality is inherently high (e.g., vision tasks) would help readers understand where these findings might or might not be beneficial.*
>
> We now discuss some practical implications (see answers to other reviewers).
>
> *3. Limited Exploration of Related Work in Neural Network Verification: While the paper provides foundational complexity results, the treatment of related verification research could be expanded. There has been recent work on heuristics and approximation methods for ReLU network verification, which could provide important context here. Reviewing recent approaches to approximate verification or discussing alternative methods that offer partial or probabilistic verification could provide readers with additional avenues to explore beyond purely theoretical boundaries. Specifically, citing and discussing recent advancements in heuristic-based and constraint programming approaches to verification could frame the authors’ results within the current landscape of neural network verification tools and methods. This contextualization would help readers understand both the novelty and the limitations of purely theoretical approaches in applied contexts.*
>
> We extended our discussion of related work (see also answers to other reviewers).
>
> *The relationship between zonotope containment and ReLU network surjectivity is intriguing but could be more accessible. Could you provide a more intuitive explanation or visualization of this connection? Could you also provide illustrative examples of injective/surjective/bijective ReLU networks?*
>
> We have added further clarification to make this connection more accessible. But we believe that the mentioned aspects are hard to illustrate properly and refrain from doing so.
>
> *Given the coNP-hardness of injectivity and verification, how should practitioners approach network design to avoid infeasibility? Are there specific heuristics or guidelines that could help manage these complexities?*
>
> Our reductions do not directly guide towards efficiently solvable cases or heuristics.
>
> *The paper focuses on theoretical hardness but doesn’t discuss how existing heuristic or approximate verification methods could be employed in light of your results [1-3]. Could you comment on where your results might influence the development or improvement of such methods?*
>
> Our hardness results do not show how to improve certain heuristics. They rather show their limits.

---

> ### Author Response · Authors · 2024-11-22
> **2/2**
>
> *Neural network verification is a computationally demanding process and has been shown to be NP-hard even for simple neural networks with a single hidden layer [1]. Could the authors clarify the novel contributions of your coNP-hardness result compared to existing NP-hardness results. Specifically, how your result provides new insights or implications for verification techniques that weren't apparent from previous hardness results?*
>
>   Our hardness result hols for arbitrary input sets while previous results used input sets specifically tailored towards certain networks. Thus, we proved that it does not help to consider restricted input sets in order to become tractable. The intractability already stems from the network alone. (See also answers to other reviewers.)
>
> *Could you provide more insight into the chosen parameterization for injectivity (in terms of input dimension) and its realistic applicability? For example, how might this approach scale in contexts where input dimensionality is inherently high, such as in image recognition (e.g., MNIST images vs. high-resolution images)?*
>
> In applications with a high input dimension the parameterized FPT-algorithm clearly is not feasible.
> Nevertheless, it is still much faster than the naive XP-algorithm and is a theoretical classification result.
>
> *Given the exponential runtime for injectivity checking based on input dimension, is there potential for further optimization or approaches for improving the algorithm's efficiency? Additionally, could you discuss any possible conjecture about the lower bounds that might limit algorithmic improvement?*
>
> In Corollary 7, we provide a lower bound based on the Exponenital Time Hypothesis stating that ReLU-Layer Injectivity cannot be solved in~$2^{o(m+d)}$ time, unless the ETH fails. Improving the running time for our FPT-algorithm might be possible. Note that we pose this as an open question in the conclusion.

---

> ### Comment · Reviewer_gZ12 · 2024-11-22
>
> I appreciate the authors’ responses. However, most of my concerns remain unaddressed:
>
> 1. Without concrete analyses or examples demonstrating the injectivity of neural networks, it is difficult to grasp the likelihood of a neural network (e.g., a random one) being injective.
> 2. As noted by Reviewer A44m, a function mapping from a high-dimensional space to a lower-dimensional one cannot be injective, which is typically the case in deep learning applications. Thus, verifying the injectivity of neural networks may hold limited practical relevance.
> 3. The NP-hardness of neural network verification has been extensively studied, along with efficient and effective methods. The coNP-hardness of injectivity verification, which indirectly suggests the hardness of neural network verification, does not provide additional insights into this problem.
> 4. While this work highlights the limitations of existing verification methods (as stated in the rebuttal), it does not contribute to improving them. This is particularly puzzling, given that current practical methods have proven effective in a wide range of neural network verification applications. Merely demonstrating theoretical limitations in a worst-case scenario does not seem particularly meaningful.
>
> Based on these considerations, I have decided to lower my score to 3 and recommend rejecting the paper.

---

> > ### Author Response · Authors · 2024-11-25
> >
> > We thank the reviewer for the anwser.
> >
> > 1.  A layer map is injective if and only if the linear map on every cone of the polyhedral subdivision has full rank as observed by Puthwala et al.  Note that this is also the characterization of injectivity that we use for the hardness result as well as the FPT algorithm. For a concrete example, see, e.g., the anwser to Reviewer A44m. Moreover, Puthwala et al. describe a procedure to create injective ReLU neural networks. Nevertheless, we do not see how a concrete example of an injective layer helps in estimating the likelihood of being injective.
> >
> > 2. See answer to Reviewer A44m on the importance of injectivity. The particular impact of our result was already explicitely acknowledged by you in your first review.
> >
> > 3. We do not see how the hardness of injectivity implies the hardness of verification (also not indirectly) since these are different problems. Could you please specify what you mean? Moreover, in your first review you explicitely acknowledge that our hardness result, independent of specific input conditions, is an insightful addition and sets the work apart from prior studies.
> >
> > 4. Understanding theoretical limitations and worst cases is generally meaningful and provides valuable insights. As you also wrote in your first review, we show interesting connections to computational geometry and zonotope problems and also give an FPT-algorithm with potential for practical improvements.

---

### Author Response · Authors · 2024-11-22
**General comment**

The main criticism of the reviewers consisted of the following points:
- Additional related work.
- Some polyhedral and graph-theoretical terminology that we used was not defined in the paper.
- We could have provided more intuition throughout the paper.

 In response to the reviewers' feedback, we have made the following revisions to the paper, taking all these comments into account:

- The related work section has been thoroughly updated. We also enhanced the "our contribution" paragraph to position our results more precisely within the existing literature, providing a clearer comparison to previous work.
- We have ensured that all terminology used in the paper is now clearly defined, making the paper fully self-contained and accessible to readers without prior knowledge of specific terms in polyhedral geometry and graph theory.
- To address concerns about the lack of intuition, we have refined the proof sketches and, in particular, added illustrations to provide a clearer understanding of key concepts, making the overall presentation more intuitive and accessible.

---

### Meta-Review · Area_Chair_iUGa · 2024-12-16

**Metareview:**

The paper investigates the computational challenges and theoretical limitations associated with key properties of ReLU neural networks: injectivity, verification, and surjectivity. The authors had tried to address the reviewers' comments and suggestions. However, some reviewers still think that their concerns have not addressed properly. The reviewer pointed out that without concrete analyses or examples demonstrating the injectivity of neural networks, it is difficult to grasp the likelihood of a neural network (e.g., a random one) being injective. Unfortunately, some reviewers still do not support the acceptance of the paper. Based on the reviews and recommendations, I cannot accept the paper at the current stage. Although the paper is not accepted this time, I strongly encourage the authors to resubmit it to the next conference venue.

**Additional Comments On Reviewer Discussion:**

NA

---

### Decision · Program_Chairs · 2025-01-22

Reject